# PerturbNet predicts single-cell responses to unseen chemical and genetic perturbations

Hengshi Yu [1,4], Weizhou Qian [2,4], Yuxuan Song [2] & Joshua D Welch [2,3 ✉]

## Abstract

Chemical and genetic perturbations, such as those induced by small molecules and CRISPR, effect complex changes in the molecular states of cells. Despite advances in high-throughput single-cell perturbation screening technology, the space of possible perturbations is far too large to measure exhaustively. Here, we introduce PerturbNet, a flexible deep generative model designed to predict the distribution of cell states induced by unseen chemical or genetic perturbations. PerturbNet accurately predicts gene expression changes in response to unseen small molecules based on their chemical structures while also accounting for key covariates such as dosage and cell type. Moreover, PerturbNet accurately predicts the distribution of single-cell gene expression states following CRISPR activation or CRISPR interference by leveraging gene functional annotations. Our approach significantly outperforms previous methods, particularly for predicting the effects of perturbing completely unseen genes. Finally, we demonstrate for the first time that amino acid sequence embeddings can be used to predict gene expression changes induced by missense mutations. We use PerturbNet to predict the effects of all point mutations in GATA1 and nominate variants that significantly impact the cell state distribution of human hematopoietic stem cells. Using a crystal structure of GATA1 bound to DNA, we validate that these large-effect variants occur in the core DNA-contact region of GATA1 and tend to involve large changes in amino acid side-chain volume.

**Keywords** Deep Generative Model; Genome Editing; High-throughput Screening; Perturbation Prediction; Single-cell Transcriptomics
**Subject Categories** Chromatin, Transcription & Genomics; Computational Biology; Methods & Resources

## Introduction

Recent experimental developments have enabled high-throughput single-cell molecular measurement of responses to drug treatment. A high-throughput chemical screen experiment exposes cells to many distinct treatments, allowing measurement of the transcriptional or morphological effects of each treatment (Gehring et al, 2020; Srivatsan et al, 2020). Understanding how drugs influence cellular responses helps discover treatments with desired effects, potentially benefiting a myriad of therapeutic applications.

Unlike chemical perturbations, whose direct gene targets are generally unknown, genetic perturbations are designed to directly knock out, knock down, or activate one or multiple target genes. The activation or repression of these genes will not only influence their own expression but also impact other genes through a complex network of downstream gene regulatory interactions. The Clustered Regularly Interspaced Short Palindromic Repeats (CRISPR) technology allows precise design of genetic mutants through genome editing (Doudna et al, 2014). More recently, CRISPR has been combined with transcriptional activators (CRISPRa) or repressors (CRISPRi) tethered to a deactivated version of the Cas9 protein (dCas9) to enable activation or inhibition of target genes. Perturb-seq, CROP-seq, and related technologies combine CRISPR/Cas9 and single-cell RNA-sequencing (scRNA-seq) to measure single-cell responses to pooled CRISPR guide RNA libraries (Dixit et al, 2016). These techniques measure cellular responses at single-cell resolution, revealing how cell states are impacted by genetic perturbations, and have been utilized for many biomedical applications (Wang et al, 2015; Adamson et al, 2016; Datlinger et al, 2017; Ursu et al, 2022; Jin et al, 2020).

Though these technology advances have enhanced the scalability of perturbation screens, the perturbation space is far too large to explore exhaustively. For example, the number of possible drug-like molecules is estimated to exceed $10^{60}$ (Reymond, 2015). Similarly, there are more than 200 million combinations of human single and double-gene perturbations alone, and higher-order combinations are also of interest. In fact, this combinatorial explosion is even worse due to many possible drug doses, guide RNAs, cell types and states, and drug or gene combinations.

In short, it is still not (and likely never will be) feasible to test all possible perturbations experimentally. Therefore, computational approaches that leverage sparse sampling of perturbation space to predict cellular responses to perturbation are essential.

Several recent methods have been developed to model single-cell perturbation effects. The variational autoencoder scGen (2019) predicted single-cell data from novel combinations of treatment and cell type using latent space vector arithmetic (Lotfollahi et al,

[1]Department of Biostatistics, University of Michigan, Ann Arbor, MI 48109, USA. [2]Department of Computational Medicine and Bioinformatics, University of Michigan, Ann Arbor, MI 48109, USA. [3]Department of Computer Science and Engineering, University of Michigan, Ann Arbor, MI 48109, USA. [4]These authors contributed equally: Hengshi Yu, Weizhou Qian. ✉E-mail: welchjd@umich.edu

2019). In 2020, Lotfollahi et al introduced a conditional variational autoencoder framework, balancing representations under two treatment conditions using a similarity score based on counterfactual inference (Johansson et al, 2016; Lotfollahi et al, 2020). Burkhardt et al identified perturbation effects across the cellular manifold with graph signal processing tools (2021) (Burkhardt et al, 2021). Yeo et al proposed a generative model using a diffusion process over a potential energy landscape to learn the underlying differentiation landscape from time-series scRNA-seq data and to predict cellular trajectories under perturbations (2021) (Yeo et al, 2021). The Compositional Perturbation Autoencoder (CPA) framework (2021) generated single-cell data under new combinations of observed perturbations using latent space vector arithmetic (Lotfollahi et al, 2023. In 2022, chemCPA was introduced, equipped with transfer learning to outperform CPA in predicting unseen drug effects (Hetzel et al, 2022; Lotfollahi et al, 2023). In 2023, GEARS (Roohani et al, 2024), a deep learning method integrated with knowledge graphs, was developed to model genetic perturbation effects. Most recently, Biolord (2024) was introduced as a deep generative model that learns disentangled representations in single-cell data, enabling the prediction of effects from both chemical and genetic perturbations (Piran et al, 2024).

One key limitation of many existing approaches is their focus on predicting only new combinations of treatments and/or cell types, which prevents them from predicting the effects of entirely unseen perturbations. Furthermore, these methods often assume that cell state is independent of perturbation, making it difficult to accurately predict perturbations that selectively promote or inhibit specific cell states. Most recent approaches, like GEARS, are typically designed for a specific type of perturbation, limiting their generalizability to other perturbation types such as variants introduced to the genome. Finally, these methods primarily predict the mean gene expression changes induced by perturbations, overlooking valuable information contained within the distribution of cell states.

Recent studies have shown that genetic perturbations can induce shifts in cell state, causing the cells to preferentially occupy certain cell states while disfavoring others. For example, Norman et al (Norman et al, 2019) observed that CRISPR activation (CRISPRa) of distinct pairs of genes in K562 cells induced some cells to shift toward erythroid, granulocyte, or megakaryocyte-like states or arrest cell division. Ursu et al found that inducing missense mutations in KRAS and TP53 caused "a functional gradient of states" in A549 cells (Ursu et al, 2022). Similarly, single-nucleotide variants in GATA1 induced cell differentiation shifts in human hematopoietic stem cells (Martin-Rufino et al, 2023). That is, in a particular tissue under homeostatic conditions, there is a wild-type distribution of cellular gene expression states $p(X)$. Treating cells with a perturbation $G$ changes their cell state distribution to some new $p(X|G)$.

Inspired by these findings and to address the limitations of previous approaches, we propose PerturbNet, a novel and flexible framework that can sample from the distribution of cell states given only the features of a new perturbation. PerturbNet bridges the space of drug treatments or genetic perturbations with cell states using a conditional normalizing flow (Papamakarios et al, 2021), enabling translation between the perturbation and cell state domains (Baltrusaitis et al, 2018). This framework is broadly applicable to high-throughput measurements of drug treatments or genetic perturbations. Crucially, PerturbNet makes distributional predictions for both observed and unseen treatments.

We demonstrate that PerturbNet effectively predicts the distribution of gene expression profiles induced by various types of perturbations and outperforms existing methods. Notably, PerturbNet can predict the effects of missense mutations on global gene expression, a type of perturbation not addressed by previous approaches. To showcase its practical utility, we applied PerturbNet to novel GATA1 mutations as an example, revealing key associations between variant properties and their effects. We further enhance PerturbNet's interpretability by identifying key perturbation features that influence cell state distributions. Overall, PerturbNet, combined with downstream analysis, provides valuable insights for designing and interpreting perturbation experiments.

## Results

### PerturbNet maps perturbation representations to cell states

PerturbNet consists of three neural networks: a perturbation representation network, a cellular representation network, and a network that maps from perturbations to cell states (Fig. 1A). The representation networks are first trained separately on large numbers of perturbations and cell profiles. Then the mapping network uses high-throughput perturbation response data such as Perturb-seq, in which both perturbation and cell states are observed, to learn a continuous mapping between the space of possible perturbations and the space of possible cell states. The intuition behind our approach is that perturbations and cells each have an underlying structure—that is, they lie in some low-dimensional space—and the effect that a perturbation exerts on cell state is given by some unknown function that maps between the two spaces (Fig. 1B). The mapping from perturbations to cell states is highly complex and not one-to-one; cells may exist in many states after a particular perturbation, and distinct perturbations may induce similar cell state distributions. To capture such complex relationships, we implement the mapping network with a conditional invertible neural network (cINN), which fits a conditional normalizing flow and can approximate arbitrary conditional distributions. Our approach is inspired by the idea of network-to-network translation, which has been used to generate images conditioned on text descriptions (Rombach et al, 2020). We confirmed that the losses for each component of PerturbNet converged for each dataset (Appendix Fig. S1).

After training, PerturbNet can make predictions about the cell states induced by a new perturbation. To do this, a description of the perturbation—such as the chemical structure of a small molecule or the identities of the genes knocked out—is first encoded into the perturbation space. Then, this location in the perturbation space is fed into the mapping network, whose output gives the distribution of locations in the cell state space induced by the perturbation. These latent cell representations can subsequently be decoded into high-dimensional gene expression levels to predict the perturbation responses of individual genes.

The PerturbNet framework has several key advantages. First, the perturbation and cell representation networks are fully modular, allowing a variety of architectures to be used depending on the data

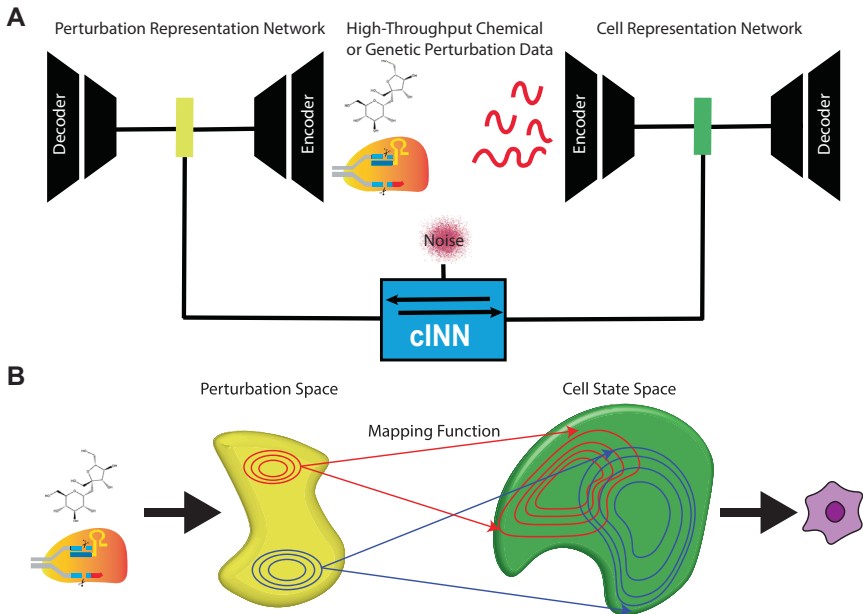

**Figure 1. PerturbNet maps perturbation representations to cell states.**

(A) PerturbNet uses two neural networks separately trained to encode large numbers of chemical or genetic perturbations (left) and cell profiles (right) into latent spaces. A conditional invertible neural network (cINN) learns to map points in perturbation space to cell state space using high-throughput measurements of perturbation effect. (B) PerturbNet can then predict the gene expression changes induced by an unseen perturbation by encoding the perturbation, passing its representation through the cINN, and decoding the resulting cell states.

type. For example, we can use convolutional and recurrent networks for representing small-molecule structures, multilayer perceptrons for gene expression data, or transformer architectures for sequence data. A second and related advantage is that we can "mix and match" the same perturbation and cell representation networks in different ways; for example, after training a network that effectively represents cellular gene expression states, we can combine this same network with a representation network for either small molecules or genetic perturbations without having to re-train the cell representation network. An additional advantage is that the representation networks can be pretrained on unpaired perturbation and cell observations, which are usually available in much larger quantities than the paired perturbation response data. If a high-quality pretrained model already exists for encoding a particular type of data, we can directly plug it into the PerturbNet without any further training. The cINN architecture used for the mapping network confers several advantages, including stable and efficient training and a mapping that is invertible by construction (see "Methods" for details). In addition, the mapping network can model additional covariates when available, such as dose or cell type, by training the mapping function using both the perturbation representation and the covariates.

In this paper, we analyzed three different types of perturbation data: genetic perturbation using CRISPRi or CRISPRa; chemical perturbations; and genetic perturbation using CRISPR genome editing of protein-coding sequences. Each type of perturbation data requires a different type of perturbation representation network. To represent CRISPRi or CRISPRa perturbations, we developed a new multilayer perceptron variational autoencoder that represents gene combinations using gene ontology terms. We represented chemical

perturbations using the previously published ChemicalVAE architecture (Gómez-Bombarelli et al, 2018), which takes string representations as input. For coding sequence perturbations, we obtained latent representations from the ESM transformer (Rives et al, 2021). For cell representation networks, we used a variational autoencoder with either Gaussian (for normalized data) or negative binomial (count data) likelihood (see "Methods" for details).

## PerturbNet predicts response to unseen small-molecule treatments

We first investigated whether PerturbNet can predict response to unseen drug treatments. Because the pharmacological properties of a small molecule are related to its chemical structure, we converted molecules into simplified molecular-input line-entry system (SMILES) format (Weininger, 1988), which represents molecular structures as character strings. Then we adopted a published architecture named ChemicalVAE (Gómez-Bombarelli et al, 2018), which takes the one-hot encoded SMILES as input and can provide low-dimensional latent representations for molecules (Fig. 2A). We pretrained the ChemicalVAE on the ZINC dataset (Irwin et al, 2005), which contains 250,000 molecules. Such a large dataset can significantly improve the generalizability of the ChemicalVAE in representing unseen drugs. In addition, the modular design of PerturbNet allowed us to utilize the same pretrained ChemicalVAE model across all datasets in the subsequent tasks.

We then trained two different cell representation networks on the gene expressions profiles from LINCS dataset (Subramanian et al, 2017) and the sci-Plex (Srivatsan et al, 2020). The LINCS dataset consists of 677,159 microarray measurements from 170

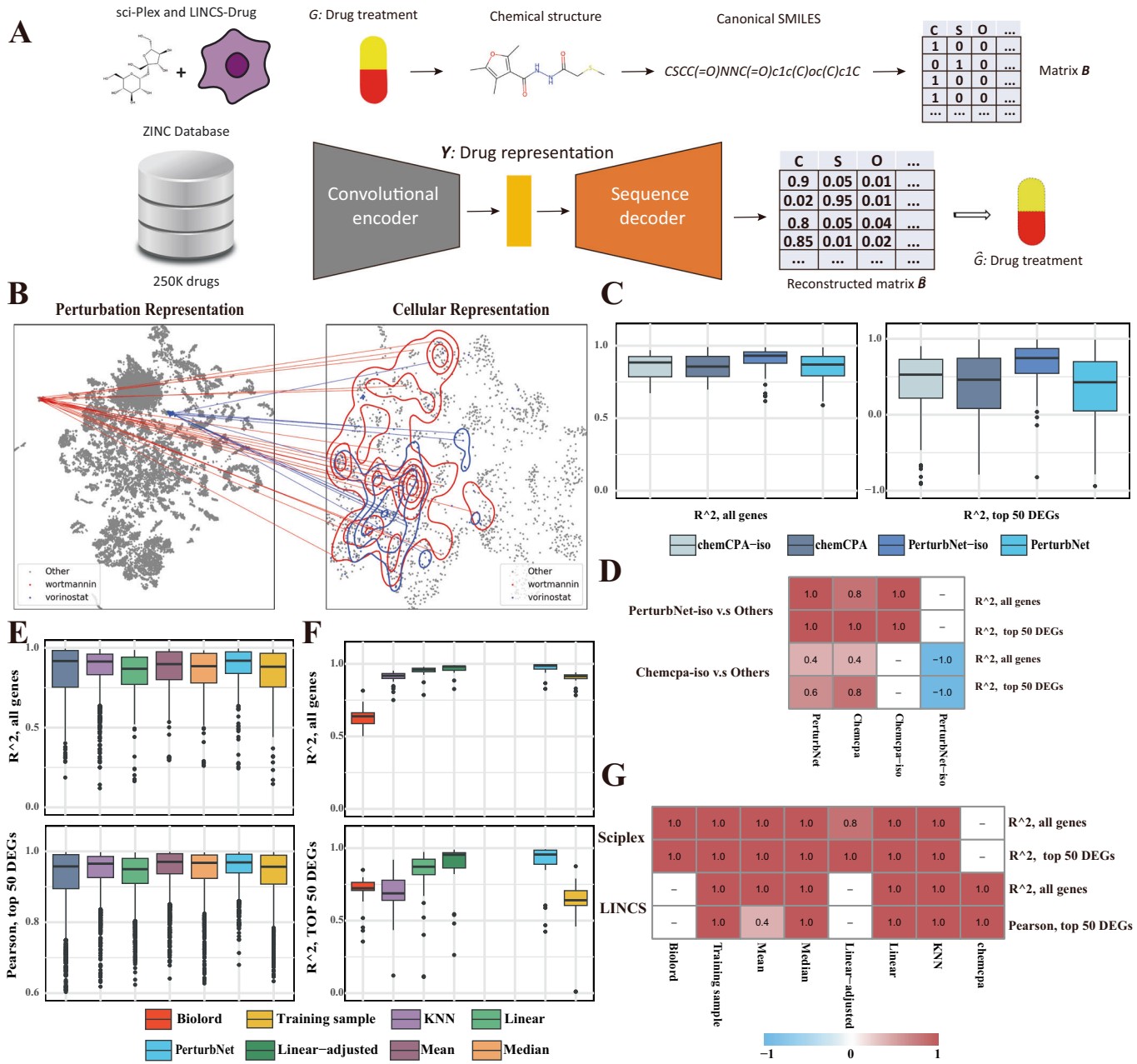

**Figure 2. PerturbNet predicts response to small-molecule treatment.**

(**A**) Diagram of the chemical variational autoencoder (ChemicalVAE) architecture for encoding small molecules represented as SMILES strings. The network was trained on the ZINC dataset, consisting of approximately 250,000 drug-like molecules. No gene expression information is used for training the drug representation network. Note that a cell representation network is also trained on gene expression values (not shown here). (**B**) Visualization of PerturbNet predictions for two distinct perturbations from LINCS-Drug dataset. The UMAP coordinates are computed from the latent spaces of the perturbation network (left) and cell state network (right). The mapping function learned by the cINN is indicated with lines connecting the perturbation and cell state representations. The predicted cell state distributions are also indicated with contour lines. (**C**) Box plots of $R^2$ calculated on all genes and top 50 differentially expressed genes (DEGs) in one test split of LINCS-Drug dataset for chemCPA and PerturbNet with (-iso) and without stereoisomers shared between train and test sets ($n = 100$, plots for additional splits are shown in the supplement). (**D**) Heatmap showing the proportion of train/test splits in which PerturbNet or chemCPA (trained with stereoisomers) outperforms the same model trained without stereoisomers across five test splits of the LINCS-Drug dataset. "Outperforms" here means a significant difference by a one-sided Wilcoxon test. (**E**) Box plots of evaluation metrics in one test split of LINCS-Drug dataset ($n = 1889$, plots for additional splits are shown in the supplement). (**F**) Box plots of evaluation metrics in one test split of sci-Plex dataset ($n = 38$, plots for additional splits are shown in the supplement). The y axis is truncated at 0. (**G**) Heatmap showing the proportion of train/test splits in which PerturbNet achieves significantly better performance (based on the one-sided Wilcoxon tests) compared to competing models (columns) across five train/test splits of sci-Plex and LINCS-Drug dataset. Note: for all box plots in this panel, the box plots show the median (center line), the 25th and 75th percentiles (box bounds), and 1.5× the interquartile range (whiskers). Points beyond the whiskers are plotted as outliers. Source data are available online for this figure.

different cell lines treated by 19,990 compounds. The sci-Plex dataset contains 648,857 scRNA-seq profiles from three cell lines treated with 180 compounds.

Finally, we trained cINNs for both the LINCS and sci-Plex datasets to create mappings from the ChemicalVAE latent space to the cell state space. After training the three modules, PerturbNet can generate and translate informative representations for both perturbations and cell states. For example, PerturbNet predicted the effect of two distinct drugs on similar cell types (Appendix Fig. S2) in the LINCS-Drug dataset (Fig. 2B). We observed clear data structures, such as clusters of similar perturbations and cell states, in the UMAP of the latent representations. Additionally, the predicted cellular distributions for the two distinct drugs exhibit significant differences (Fig. 2B).

To quantitatively evaluate the performance of PerturbNet in predicting the effects of unseen small-molecule treatments, we created five different train/test splits for the LINCS and sci-Plex datasets. We compared against multiple baseline models: training samples, a linear model, training data mean and median (Kernfeld et al, 2023), and $k$-nearest neighbors (KNN). The training sample model selects cells from all treatments observed during training. This comparison is crucial, as perturbations with minimal or no effect can, in principle, be predicted accurately by simply guessing the mean response of all cells. The linear model predicts cellular responses using ChemicalVAE embeddings as predictors. This serves as another naive baseline, as the relationship between drug structures and gene expression is most likely non-linear. The KNN model selects cells based on the 5 nearest neighbors ($k = 5$) in the perturbation space, which is a smarter baseline model. Finally, we compared against previous approaches for this task, including chemCPA and Biolord. We included the models with the best reported performance for each benchmark. Specifically, we compared PerturbNet to chemCPA on the LINCS-Drug dataset, as the data are normalized and compared PerturbNet to the Biolord on sci-Plex raw count data.

We noticed that the majority of treatments in the LINCS-Drug dataset are stereoisomers–the molecules share the same molecular formula and arrangement of atoms but differ in their precise 3D orientations. These stereoisomers, though sharing a chemical formula, can result in either similar or different effects. However, some previous computational methods do not distinguish stereoisomers, giving them the same representation. For example, chemCPA uses RDkit, which does not distinguish between stereoisomers, whereas the chemical representation used by PerturbNet is different for stereoisomers. Furthermore, including the stereoisomers in training and test sets may cause data leakage and boost the model's performance if many stereoisomers have similar effects on gene expression. Therefore, we investigated how stereoisomers affect the models' performance by training both chemCPA and PerturbNet either with or without stereoisomers shared between the training and test sets. Our results show that both models achieve higher $R^2$ when trained with stereoisomers of the test set drugs (Figs. 2C and EV1A; Table 1). In addition, Fig. 2D shows the fraction of train/test splits for which PerturbNet or chemCPA trained with stereoisomers significantly outperforms the same model trained without stereoisomers (based on one-sided Wilcoxon tests). Both models exhibit positive ratios when compared to their non-stereoisomer versions, indicating that models trained with stereoisomers perform significantly better. These results indicate that caution is needed when the training data

**Table 1.** Summary of mean and median evaluation metrics on the LINCS-Drug dataset for chemCPA and PerturbNet with (-iso) and without stereoisomers shared between train and test sets.

| Model | Median $R^2$ | Mean $R^2$ | Median $R^2$ (DEG) | Mean $R^2$ (DEG) |
|---|---|---|---|---|
| chemCPA-iso | <u>0.891</u> | <u>0.319</u> | <u>0.563</u> | <u>0.319</u> |
| chemCPA | 0.857 | 0.192 | 0.418 | 0.192 |
| PerturbNet-iso | **0.933** | **0.623** | **0.794** | **0.623** |
| PerturbNet | 0.874 | 0.154 | 0.394 | 0.154 |

The top-performing values are bolded and the second-best are underlined for each metric.

contains many stereoisomers. Notably, even though PerturbNet encodes stereoisomers into different embeddings, being trained on stereoisomers still brings obvious advantages. One explanation is that the stereoisomers are close neighbors in perturbation space, and stereoisomers in LINCS-Drug are more likely to have very similar effects on gene expression compared to non-isomeric compounds.

Since stereoisomers may introduce data leakage, leading to unfair comparisons, we opted to remove them and retain only unique molecules in the LINCS-Drugs dataset prior to benchmarking. We generated five distinct train/test splits, each containing 2000 unseen perturbations in the test set. We demonstrate that PerturbNet outperforms the baseline models and chemCPA, achieving the highest $R^2$ values and Pearson correlations when predicting the effects of unseen chemical perturbations on cell distributions (Table 2). These metrics were evaluated for all genes as well as the top 50 differentially expressed genes (DEGs) across all five data splits in the LINCS-Drug dataset (Figs. 2E and EV1B). The last two rows of Fig. 2G show that PerturbNet significantly outperforms the other models across multiple train/test splits.

Given that covariates such as cell types and drug dosage are crucial predictors of cell perturbation response (Appendix Supplementary Information A.1, Appendix Fig. S3), we benchmarked covariate-adjusted models using the sci-Plex dataset. During training, both the perturbations and their associated dosage and cell-type information were provided to both Biolord and Perturb-Net. PerturbNet still achieved the best performance in predicting unseen chemical perturbations with the highest mean and median $R^2$ evaluated across five train/test splits (Table 3; Figs. 2F and EV1C). Figure 2G further shows that PerturbNet consistently outperformed all the baselines and Biolord, achieving performance that is significantly better in at least four out of five train/test splits.

We observed dramatic performance differences in the mean and median baseline models between the LINCS-Drug and sci-Plex datasets. One possible reason is that LINCS-Drug comprises multi-step normalized and z-scaled microarray data, making it far less sparse than the scRNA-seq data, which only undergoes library size normalization and log1p transformation. It is also possible that the effects of LINCS-drug treatments are not as strong as those observed in the sci-Plex dataset.

## PerturbNet predicts response to unseen genetic perturbations

We next extended the PerturbNet framework to genetic perturbations by constructing an autoencoder for the target genes in

**Table 2. Summary of mean and median evaluation metrics on the LINCS-Drug dataset.**

| Model | Median $R^2$ | Mean $R^2$ | Median Pearson (DEG) | Mean Pearson (DEG) |
|---|---|---|---|---|
| Chemcpa | 0.899 | 0.862 | 0.956 | 0.931 |
| KNN | <u>0.912</u> | 0.873 | 0.965 | 0.940 |
| Linear | 0.869 | 0.847 | 0.950 | 0.932 |
| Mean | 0.901 | <u>0.881</u> | **0.971** | <u>0.954</u> |
| Median | 0.888 | 0.866 | <u>0.968</u> | 0.944 |
| PerturbNet | **0.919** | **0.894** | **0.971** | **0.958** |
| Training sample | 0.887 | 0.853 | 0.958 | 0.936 |

The top-performing values are bolded, and the second-best are underlined for each metric.

**Table 3. Summary of mean and median evaluation metrics on the sci-Plex dataset.**

| Model | Median $R^2$ | Mean $R^2$ | Median $R^2$ (DEG) | Mean $R^2$ (DEG) |
|---|---|---|---|---|
| Biolord | 0.587 | 0.554 | 0.716 | 0.658 |
| KNN | 0.918 | 0.893 | 0.664 | 0.328 |
| Linear | 0.954 | 0.936 | 0.825 | 0.569 |
| Linear-adjusted | <u>0.976</u> | <u>0.964</u> | <u>0.917</u> | <u>0.824</u> |
| Mean | −6.271 | −6.156 | −10.161 | −11.838 |
| Median | −0.636 | −0.647 | −2.296 | −2.364 |
| PerturbNet | **0.984** | **0.968** | **0.951** | **0.865** |
| Training sample | 0.915 | 0.894 | 0.596 | 0.356 |

The top-performing values are bolded, and the second-best are underlined for each metric.

combinatorial genetic perturbations. Genome editing with CRISPR/Cas9 directly modifies the DNA sequence, leading to changes in the protein-coding sequence or non-coding regulatory sequence. In contrast, genetic perturbations using CRISPR activation (CRISPRa) or CRISPR interference (CRISPRi) do not change the original DNA sequence, but rather change gene expression (Norman et al, 2019). These types of perturbations can be described by the identities of their target genes (Dixit et al, 2016). To encode such genetic perturbations, we developed a new autoencoder that we call GenotypeVAE (Fig. 3A). Our key insight is that the numerous functional annotations of each gene (organized into a hierarchy in the gene ontology) provide features for learning a low-dimensional representation of both individual genes and groups of genes. The Gene Ontology Consortium has annotated 18,832 human genes with a total of 15,988 terms (after filtering to remove terms with very low frequency). Using these annotations, we can describe each target gene $g$ as a one-hot vector of length 15,988, where a value of 1 in the vector element corresponding to a particular term indicates that the gene has that annotation. If we have a genetic perturbation with multiple target genes, we can simply take the union of the GO annotations from all of the perturbed genes.

We trained a variational autoencoder with fully connected layers to reconstruct these binary annotation vectors from a latent representation. Our approach is inspired by a previous study that used neural networks to embed genes into a latent space based on

their gene ontology annotations (Chicco et al, 2014). We trained GenotypeVAE using one-hot representations of many possible genetic perturbations. Considering all single- and double-gene combinations of the 18,832 human genes with GO term annotations, there are ~177 million possible training data points (Fig. 3A).

We then evaluated our approach against the baseline and other existing approaches, including Biolord and GEARS, on the dataset from Norman et al. It is a CRISPRa screen with scRNA-seq data in K562 cells with 230 perturbations (Norman et al, 2019). We created five separate train/test splits for model training and evaluation.

One advantage of PerturbNet is its ability to directly model raw counts from scRNA-seq data by using a variational autoencoder (VAE) with a ZINB likelihood (Lopez et al, 2018). As shown in Figs. 3B and EV2A, modeling raw counts from scRNA-seq data enhances PerturbNet's predictions on unseen genetic perturbations, resulting in higher $R^2$ values across all five train/test splits. The mean and median $R^2$ values evaluated on all genes and DEGs are also improved when PerturbNet is trained on raw counts (Appendix Table S1). This suggests that modeling raw counts captures the inherent variability in gene expression data more effectively than normalized approaches, leading to improved generalization and predictive accuracy. Therefore, we used a VAE with ZINB likelihood for all subsequent scRNA datasets.

Table 4 presents the benchmark results on the Norman et al dataset. We evaluated PerturbNet against a linear model baseline that has been reported to outperform existing deep learning methods (Ahlmann-Eltze et al, 2024) in predicting CRISPRa and CRISPRi responses. Among all the models, PerturbNet achieves the highest mean and median $R^2$ values for both all genes and the top 50 differentially expressed genes (DEGs). When examining each split individually, PerturbNet consistently outperforms both GEARS and the baseline models on all genes across all five splits (Figs. 3C and EV2B). However, its performance is comparable to Biolord, as two splits do not show significant differences (Fig. 3D). For the top 50 DEGs, PerturbNet significantly outperforms all other methods on most test splits, though its performance is similar to that of KNN and the linear baseline on one or two splits (Figs. 3D and EV2B). Furthermore, PerturbNet achieves higher accuracy compared to GEARS and Biolord in the most challenging tasks. The difficulty of predicting post-perturbation effects is related to the number of perturbed genes (targets) and whether the target has been previously observed (e.g., in other combined target assays). In Fig. 3E, nearly all data points favor PerturbNet over GEARS for all genes, with PerturbNet showing a clear advantage in predicting completely unseen genetic perturbations ("0/1" and "0/2") for the top 50 DEGs and "large-effect genes" (a subset of top 50 DEGs with absolute log-fold changes ≥1). We also observe similar trends when comparing PerturbNet to Biolord for predicting completely unseen genetic perturbations on DEGs and large-effect genes, though Biolord performs comparably to PerturbNet on all genes (Fig. 3F). Similar results comparing PerturbNet and the baseline models are shown in (Fig. EV3).

The study by Norman et al investigated genetic interactions between gene pairs. In the simplest cases, the combined effect of perturbing two genes is equivalent to the additive effect of perturbing each gene independently. However, many genes interact in non-additive ways. For instance, overexpressing CBL and CNN1 independently results in similar transcriptomic changes, but their combined overexpression triggers a synergistic interaction that

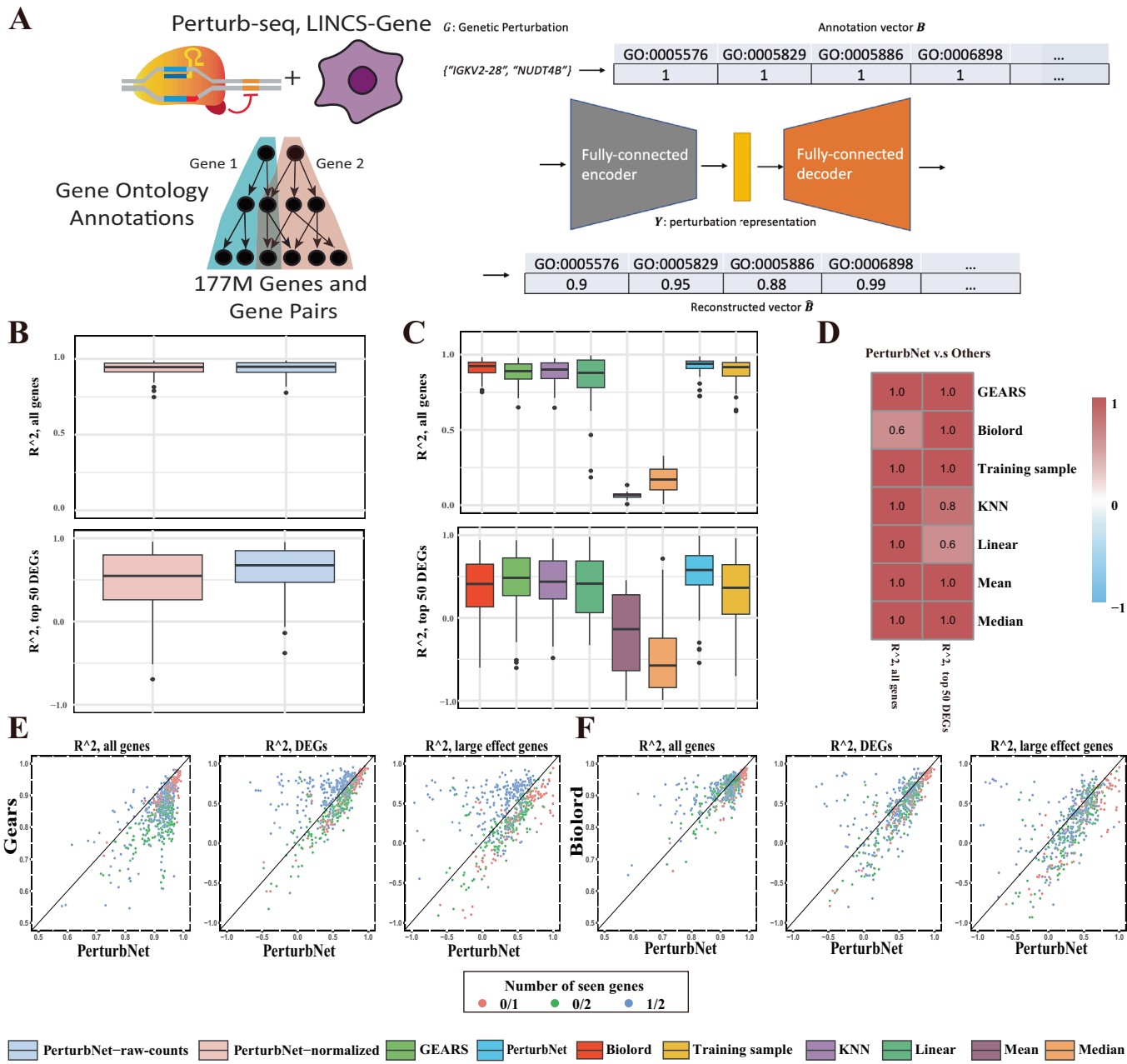

**Figure 3. PerturbNet predicts response to genetic perturbation.**

(A) Diagram of the GenotypeVAE architecture. Perturbations are represented in terms of their gene ontology annotations. The network is trained on all one- and two-gene combinations (~177 million). (B) Box plots of $R^2$ values for unseen genetic perturbations, calculated on all genes and the top 50 differentially expressed genes (DEGs) on one test split of the Norman et al dataset ($n = 33$, plots for additional splits are in the supplement). "PerturbNet-raw-count" utilizes a zero-inflated negative binomial (ZINB) likelihood for the cell representation network, while "PerturbNet-normalized" employs a Gaussian likelihood. (C) Box plots of $R^2$ values for unseen genetic perturbations, calculated on all genes and the top 50 DEGs on one test split of the Norman et al dataset. ($n = 46$, plots for additional splits are in the supplement). (D) Heatmap showing the fraction of train/test splits in which PerturbNet achieves significantly better performance (one-sided Wilcoxon tests) compared to competing models (rows) on the Norman et al dataset. (E, F) Scatter plots of $R^2$ values for unseen genetic perturbations, calculated across all genes, the top 50 differentially expressed genes (DEGs), and large-effect genes from the Norman et al dataset. Data points are aggregated from all five test splits. Different colors represent the "number of seen genes" Labels such as "0/1" indicates that the test perturbation affects one unseen gene, while "0/2" indicates that two unseen genes are perturbed. "1/2" denotes that two genes are perturbed, but one of the target effects has already been observed independently or in combination with other genetic perturbations. Note: for all box plots in this panel, the box plots show the median (center line), the 25th and 75th percentiles (box bounds), and 1.5× the interquartile range (whiskers). Points beyond the whiskers are plotted as outliers. Source data are available online for this figure.

**Table 4. Summary of mean and median evaluation metrics on the Norman et al dataset.**

| Model | Median $R^2$ | Mean $R^2$ | Median $R^2$ (DEG) | Mean $R^2$ (DEG) |
|---|---|---|---|---|
| Biolord | <u>0.936</u> | <u>0.923</u> | 0.540 | 0.436 |
| GEARS | 0.858 | 0.853 | 0.513 | 0.451 |
| KNN | 0.918 | 0.901 | 0.452 | 0.409 |
| Mean | −0.097 | −0.115 | −1.447 | −1.549 |
| Median | 0.170 | 0.152 | −1.408 | −1.439 |
| PerturbNet | **0.942** | **0.928** | **0.629** | **0.535** |
| Training sample | 0.930 | 0.912 | 0.430 | 0.375 |
| Linear | 0.891 | 0.830 | <u>0.561</u> | <u>0.489</u> |

The top-performing values are bolded, and the second-best are underlined for each metric.

amplifies the expression of hemoglobin genes. Therefore, we are interested in PerturbNet's ability to predict outcomes influenced by non-additive genetic interactions. Given that comparisons between PerturbNet and existing methods are largely dominated by the number of observed genes for unseen combined perturbations, we tested for association between several factors and both $R^2$ on all genes and $R^2$ on the top 50 DEGs. After controlling for all relevant covariates, we found that many subtypes of genetic interactions are significantly associated with the $R^2$ values, and most of the coefficients are positive (Appendix Fig. S4). This indicates that PerturbNet predicts non-additive genetic interactions more accurately than cases with no or unknown interactions. For example, in the linear model using $R^2$ on DEGs as the response, PerturbNet shows improved prediction for the neomorphic ($\beta$: 0.125, P value: $9.77 \times 10^{-3}$), potentiation ($\beta$: 0.188, P value: $8.58 \times 10^{-3}$), strong synergy and dissimilar phenotype ($\beta$: 0.228, P value: $9.08 \times 10^{-7}$), strong synergy and similar phenotype ($\beta$: 0.234, P value: $3.10 \times 10^{-7}$) and suppressors ($\beta$: 0.131, P value: $2.26 \times 10^{-3}$), while performing worse in cases of redundant ($\beta$: −0.273, P value: $1.07 \times 10^{-5}$). Overall, PerturbNet demonstrates strong predictive accuracy even when non-additive genetic interactions between previously unobserved genes are present.

## PerturbNet predicts response to coding sequence mutation

In addition to CRISPRi and CRISPRa, which do not change the DNA sequence within a cell, Perturb-seq can also be combined with CRISPR genome editing. For example, Ursu et al recently used a multiplex CRISPR screen to introduce many different coding sequence mutations into the TP53 and KRAS genes of A549 cells (Ursu et al, 2022). This study found that the genome edits caused "a functional gradient of states" with continuously varying gene expression profiles. Similarly, Martin-Rufino et al recently used CRISPR base editing with scRNA readout to identify how GATA1 mutations change hematopoietic cell differentiation (Martin-Rufino et al, 2023). Unlike CRISPRa or CRISPRi perturbations, which can be represented in terms of the identities of the target genes, genome editing perturbations are best represented as the distinct amino acid sequences of either the wild-type or edited genes.

To extend PerturbNet for predicting single-cell gene expression responses to coding sequence variants, we developed a strategy for embedding amino acid sequences. We chose to use the pretrained evolutionary scale modeling (ESM) network (Rives et al, 2021) to obtain latent representations of the unique protein sequences produced by genome editing (Fig. 4A). ESM is a previously published network that was pretrained on about 250 million protein sequences from the UniParc database (Rives et al, 2021). Unlike the chemicalVAE and the GenotypeVAE used above, the ESM encodings are deterministic for a given input sequence; thus, to avoid overfitting when training the cINN on ESM representations, we added a small amount of Gaussian noise sampled from $\mathcal{N}(\boldsymbol{0}, 0.001\boldsymbol{I})$.

We then trained PerturbNet on the Ursu dataset, which measured the effects of various mutations in the TP53 and KRAS genes (Ursu et al, 2022), as well as on the Jorge dataset, which identified functional variants involved in human hematopoiesis. The CRISPR guide RNA sequences were preprocessed to obtain a single, complete protein sequence label for each individual cell. After filtering out sequences with too few cells, we retained 163 unique protein sequences for the Ursu dataset and 257 sequences for the Jorge dataset. For both datasets, we used a VAE with ZINB likelihood to learn cell state representations.

As there are no appropriate competing methods for this task, to the best of our knowledge, we benchmarked the model against only the baseline models. Since most coding variants induce minimal shifts in cell states, the baseline models can perform surprisingly well on all genes and DEGs. However, when focusing on large-effect genes—those that experience significant changes due to perturbations—the training sample baseline fails to make accurate predictions. In the Ursu (TP53 + KRAS) dataset, PerturbNet consistently outperforms all baselines except the KNN (Figs. 4D,E and EV2C; Table 5). One possible explanation is that the two distinct protein sequences allow the KNN model to better capture the correct neighbors, as nearby coding sequence perturbations often have small but similar effects, which are accurately described by the KNN relationships. However, in the Jorge (GATA1) dataset, PerturbNet consistently achieves significantly higher $R^2$ values than all the baseline models (Figs. 4D,E and EV2C; Table 6), demonstrating its ability to accurately predict subtle effects from coding variants (note that median baseline predicts zero counts for large-effect genes and top 50 DEGs in the Jorge dataset).

We applied PerturbNet to predict gene expression changes induced by all possible point mutations in the GATA1 sequence. The workflow is illustrated in Fig. 4B. PerturbNet was trained on the Jorge dataset, utilizing all available sequences (257 unique GATA1 sequences, including the wild-type). GATA1 is a critical Zinc finger transcription factor essential for the differentiation of various blood and immune cells. The canonical GATA1 sequence consists of 413 amino acids, and each position can theoretically be mutated to one of the other 19 amino acids. Therefore, we predicted cell distributions for a total of 7902 unique protein sequences, which include all 7847 possible point mutations ($413 \times 19$) and the observed mutations. We generated 100 cells for each unique protein sequence and processed the data for downstream analysis. Interestingly, PerturbNet predicts three distinct classes of mutations: erythroid-depleted, erythroid-intermediate and erythroid-enriched. Each class of mutations shows a distinct cell state distribution during hematopoietic differentiation (Fig. 4F). Previous research by Jorge et al demonstrated that certain GATA1 mutations can block the differentiation

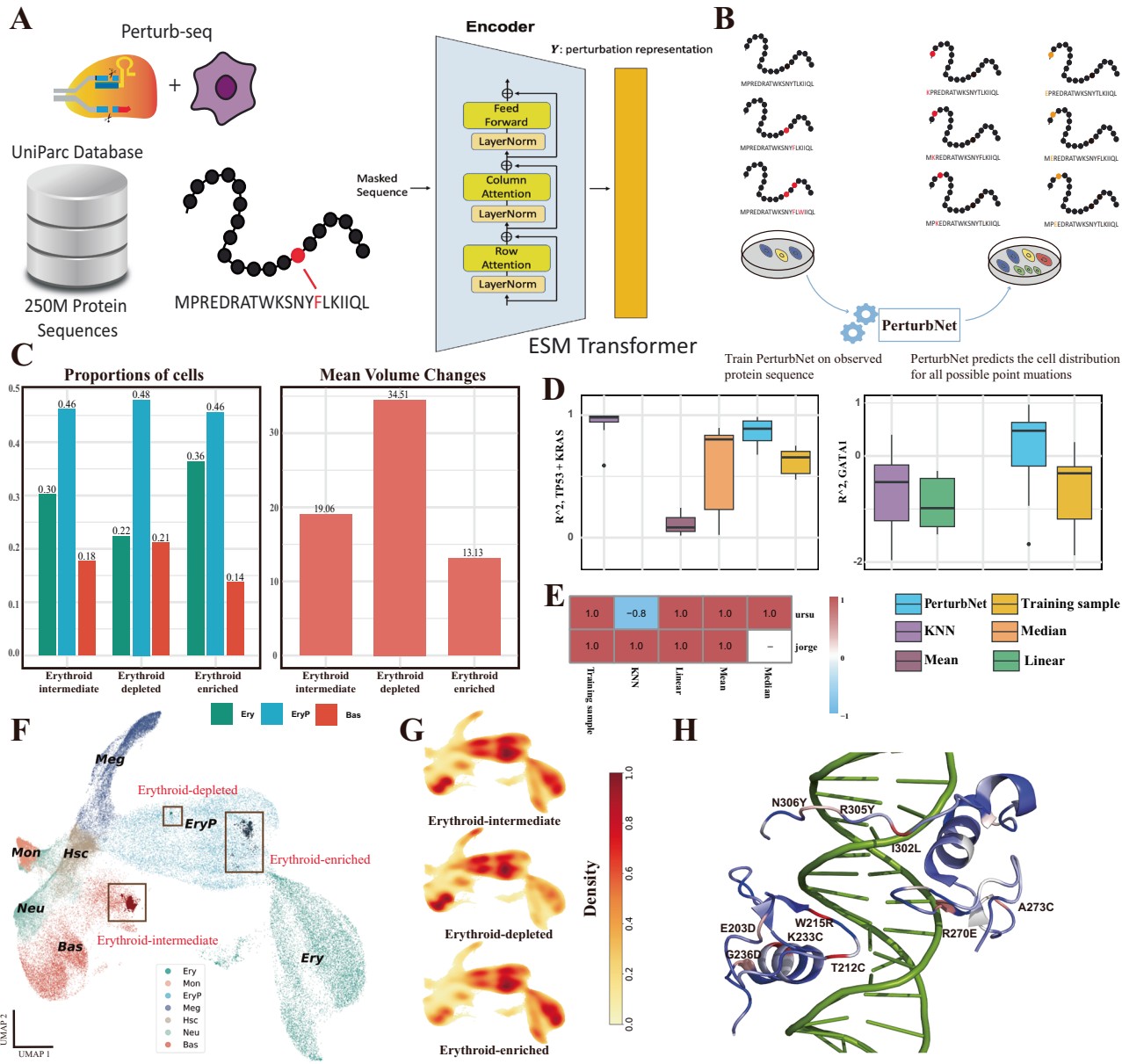

**Figure 4. PerturbNet predicts response to coding sequence mutation.**

(A) Diagram of the approach for training a representation network for coding sequence mutations. Each perturbation is an amino acid sequence edited by CRISPR. We used an evolutionary scale modeling (ESM) transformer pretrained on the UniParc database, containing 250M sequences. (B) Workflow for predicting the effects of all possible point coding variants along a specific protein sequence. PerturbNet is trained using all available coding variants and then used to generate cell distributions for every possible protein variant at each position within the protein sequence. (C) Bar plots showing the cell-type proportions (left) and mean volume changes induced by coding variants (right) for the three predicted variant clusters along the GATA1 sequences. (D) Box plots of $R^2$ values for unseen genetic perturbations (coding variants) calculated on the large-effect genes in the dataset from Ursu et al ($n = 10$, TP53 + KRAS) and Jorge et al ($n = 35$, GATA1). The y axis is truncated at 0. (E) Heatmap showing the proportion of train/test splits for which PerturbNet significantly outperforms (one-sided Wilcoxon tests) other models (columns). (F) UMAP visualization of predicted cell distributions for all possible point coding variants and observed coding variants on the GATA1 protein, with three distinct perturbation clusters highlighted. (G) Cell density of each perturbation cluster in the UMAP. (H) The top 10 predicted large-effect coding variants are located at or near critical GATA1–DNA interacting regions. Note: for all box plots in this panel, the box plots show the median (center line), the 25th and 75th percentiles (box bounds), and 1.5× the interquartile range (whiskers). Points beyond the whiskers are plotted as outliers. Source data are available online for this figure.

of erythroid progenitors into terminal erythroid cells. Similarly, we found a group of mutations (the erythroid-depleted cluster) that significantly disrupted this differentiation process. In Fig. 4C,G, the mutations from the erythroid-depleted cluster result in a markedly lower proportion of terminal erythroid cells compared to mutations from other two clusters. ($P$ values of proportional z-test are less than $2.2 \times 10^{-16}$). Thus, PerturbNet predicts that GATA1 mutations in the erythroid-depleted cluster are likely to inhibit terminal erythroid differentiation and contribute to abnormal hematopoiesis.

**Table 5. Summary of mean and median evaluation metrics on the Ursu et al dataset.**

| Model | Median $R^2$ (large) | Mean $R^2$ (large) | Median $R^2$ (DEG) | Mean $R^2$ (DEG) |
|---|---|---|---|---|
| KNN | **0.982** | **0.942** | **0.986** | **0.948** |
| Linear | −22.932 | −27.767 | −24.400 | −27.518 |
| Mean | −0.473 | −0.743 | −0.806 | −0.650 |
| Median | 0.022 | −0.006 | −0.621 | −0.691 |
| PerturbNet | <u>0.870</u> | <u>0.814</u> | <u>0.882</u> | <u>0.823</u> |
| Training sample | 0.697 | 0.647 | 0.852 | 0.787 |

The top-performing values are bolded, and the second-best are underlined for each metric.

**Table 6. Summary of mean and median evaluation metrics on the Jorge et al dataset.**

| Model | Median $R^2$ (large) | Mean $R^2$ (large) | Median $R^2$ (DEG) | Mean $R^2$ (DEG) |
|---|---|---|---|---|
| KNN | <u>−4.031</u> | **−41.873** | 0.971 | 0.955 |
| Linear | −10.186 | −139.809 | 0.837 | 0.685 |
| Mean | −68.699 | −653.178 | 0.027 | −0.208 |
| Median | NA | NA | NA | NA |
| PerturbNet | **0.2** | −99.65 | **0.987** | **0.983** |
| Training sample | −4.519 | <u>−44.121</u> | <u>0.972</u> | <u>0.958</u> |

The top-performing values are bolded, and the second-best are underlined for each metric. Because the median model predicts zero counts for large-effect genes and the top 50 DEGs, valid metrics cannot be computed for it.

In addition, we observed that the mutations in the erythroid-depleted cluster tend to involve larger volume changes in amino acid side chains compared to other mutations (Fig. 4C, right). This prompted us to investigate the biochemical factors associated with the effects of these variants. Hence, we fitted three logistic regression models to compare the probability of a mutation being assigned to one of the clusters in a pairwise comparison. The predictors included volume changes, hydrophobicity changes, and a categorical variable indicating whether the residue had any level of DNA contact. Across the three logistic regression models, larger volume changes were significantly associated with higher odds of the mutation belonging to clusters located on the left side of the UMAP plot, except in the comparison between the erythroid-depleted cluster and erythroid-intermediate cluster (Appendix Fig. S5A). These results suggest that larger volume changes increase the likelihood of a variant disrupting normal cell differentiation in human hematopoiesis, likely due to a greater chance of disrupting the DNA-binding function of GATA1.

Next, we validated our predictions by examining the experimental crystal structure of GATA1 bound to DNA. We identified the top 10 large-effect mutations by ranking the log-fold changes of large-effect genes. Notably, 8 out of these 10 mutations are located in critical regions, either in direct contact with DNA or near DNA-contact residues, based on the experimental crystal structure (Fig. 4H). These mutations have the potential to alter the GATA1–DNA interface, leading to significant impacts on the transcriptome since GATA1 is a transcription factor. This finding further validates our predictions that these mutations are likely to

have large effects on the transcriptome. Finally, we confirmed that these predictions are not simply regurgitating the positions that were mutated in the training data—4 out of 10 occur at positions not seen in the training data.

Since the GATA1–DNA complex structure lacks certain residues, and many of the included residues are in critical regions, we checked if the high number of top large-effect mutations was due to positional bias in the structure. We examined mutation effect distributions for three mutation classes, as shown in Appendix Fig. S5B. Each class is defined by the DNA-contact relationship of the mutated residue. Mutations in the "DNA-contact" and "close to DNA-contact" classes show a stronger rightward skew compared to "other residues." This indicates that the largest-effect mutations are not solely due to a higher proportion of critical residues; otherwise, we would observe similar distributions for each class.

## Discussion

In summary, PerturbNet accurately predicts a wide range of common perturbations, including chemical treatments, CRISPRa/CRISPRi, and coding variants. In benchmarks, PerturbNet consistently outperformed other methods and baseline models. Its flexible architecture allows easy extension to additional perturbation types, such as gene knockouts, which can be modeled similarly to CRISPRa/CRISPRi by adapting the GenotypeVAE within PerturbNet, or by incorporating other state-of-the-art models for compatibility. Furthermore, PerturbNet focuses on predicting post-perturbation cell state distributions rather than just the means, providing richer and more complex information. PerturbNet also enables the interpretation of key perturbation features and the discovery of biological factors driving specific cell states shift (Appendix Supplementary Information A.6, Fig. EV4), providing valuable insights that can guide experimental design.

However, several challenges remain for our method. In some cases, state-of-the-art methods including PerturbNet seem to only provide moderate performance compared to baselines. It is possible that larger perturbation datasets will widen this gap, as simpler models fail to capture the complexity of perturbation responses. In addition, PerturbNet cannot predict the effects of unseen perturbations on unseen cell types. While training the cell representation network on multiple single-cell datasets may improve generalizability across cell types, the limited availability of paired perturbation-to-cell state mappings in current published data remains a significant barrier. In addition, PerturbNet is unable to extend its predictions to unseen proteins when predicting coding variants. Another assumption made by the model is that all interventions are successful and almost nontoxic, which may not hold true in real experimental conditions, potentially reducing prediction accuracy.

Still, our results open a number of exciting future directions. One possible direction is to broaden to additional types of perturbations and responses. For example, one could try to predict the trajectories of single-cell responses after a sequence of perturbations (Bergen et al, 2020). Future work could explore integrating other state-of-the-art methods for chemical and genetic perturbations to enhance perturbation representations. Training these frameworks on larger chemical databases, such as PubChem (Kim et al, 2016), or expanding GO annotation sets to include genetic perturbations with more than two target genes could also

                                     

improve model performance. Our experiments suggest that PerturbNet's prediction accuracy for cellular responses to unseen perturbations is influenced by the number of observed perturbations available for training the cINN. To improve cINN translation for single-cell data with limited observed perturbations, transfer learning (Lotfollahi et al, 2022) could be employed to leverage cINN models trained on larger datasets, such as LINCS-Drug. Furthermore, incorporating more advanced generative models, like MichiGAN (Yu et al, 2021), into the PerturbNet framework may further enhance generation performance. If multi-omic perturbation data becomes available, adapting the cell representation networks to capture more comprehensive cell states could offer broader insights into perturbation effects. We anticipate that PerturbNet and similar approaches will contribute to the design of high-throughput perturbation experiments, better utilization of large datasets, and ultimately help identify novel chemical and genetic therapies.

# Methods

### Reagents and tools table

| Reagent/resource | Reference or source | Identifier or catalog number |
|---|---|---|
| **Experimental models** | | |
| **Recombinant DNA** | | |
| **Antibodies** | | |
| **Oligonucleotides and other sequence-based reagents** | | |
| **Chemicals, enzymes, and other reagents** | | |
| **Software** | | |
| SCANPY v1.9.1 | https://scanpy.readthedocs.io/en/stable/ | |
| PerturbNet v0.0.3 | https://github.com/welch-lab/PerturbNet/tree/main | |
| GEARS v0.0.2 | https://github.com/snap-stanford/GEARS | |
| Biolord v0.0.2 | https://github.com/nitzanlab/biolord | |
| chemCPA v1.0.0 | https://github.com/theislab/chemCPA | |
| fair-ESM v2.0.0 | https://github.com/facebookresearch/esm | |
| scvi-tools v0.7.1 | https://scvi-tools.org/ | |
| **Other** | | |

## Datasets with chemical perturbations

### ZINC

We obtained the ZINC database with 250,000 compounds (Irwin et al, 2005) from the ChemicalVAE model (https://github.com/aspuru-guzik-group/chemical_vae/tree/main/models/zinc). We transformed the compounds to canonical SMILES following the ChemicalVAE

tutorial (https://github.com/aspuru-guzik-group/chemical_vae/blob/main/examples/intro_to_chemvae.ipynb) via the RDKit package (Landrum, 2016). We also utilized the chemical elements' library from this tutorial to define the one-hot matrices of drug treatments, where we constrained the maximum length of canonical SMILES strings to be 120. The chemical elements' library contains 35 unique characters, including padding symbols for SMILES shorter than 120.

### sci-Plex

We processed the entire sci-Plex dataset(GSM4150378) (Srivatsan et al, 2020) using SCANPY (Wolf et al, 2018), resulting in a dataset consisting of 648,737 cells and 5087 genes. Of these, 634,110 cells were perturbed by a total of 188 drug treatments, with 14,627 cells remaining unperturbed. For all the datasets we preprocessed with SCANPY, we followed the steps below:

```
sc.pp.normalize_total(adata, target_sum = 1e4)
sc.pp.log1p(adata)
sc.pp.highly_variable_genes(adata,     min_-
mean=0.0125, max_mean=5, min_disp=0.5)
```

After filtering out drug treatments that were incompatible with the ChemicalVAE training set, 180 drug treatments remained. The processed dataset information is summarized in Table 7. For model benchmarking, we randomly selected 38 drug treatments as unseen perturbations and the remaining 152 as observed perturbations. The train/test splits were ~80–20.

### LINCS-Drug

We obtained the LINCS dataset (Subramanian et al, 2017) from GEO (accession ID: GSE92742). The processed dataset consists of 1,319,138 cells and 978 landmark genes. The LINCS-Drug subset includes 689,831 cells treated with 20,329 drug treatments, represented by their SMILES strings. Of these, 19,990 drug treatments had SMILES lengths under 120 characters and contained only the 35 unique characters present in the ChemicalVAE training set, including padding. Among these 19,990 drugs, 5804 did not have stereoisomers. For those drugs with stereoisomers, we retained only the structure with the largest number of associated cells, resulting in a total of 8955 unique molecules. For model benchmarking, we randomly selected 1000 perturbations from molecules without stereoisomers and 1000 perturbations from the unique stereoisomers. Five distinct train/test splits were created, maintaining an 80–20 ratio for each split. The processed dataset information is summarized in Table 7.

We transformed the SMILES strings of drug treatments of the sci-Plex and LINCS-Drug data to their one-hot matrices according to the chemical elements' library derived from the ZINC database.

## Datasets with gene activation and coding sequence mutations

### GO annotations

We obtained the GO annotation dataset for human proteins from the GO Consortium (Aleksander et al, 2023) at http://geneontology.org/docs/guide-go-evidence-codes. We removed the annotations of three sources without sufficient information: inferred from electronic annotation (IEA), no biological data available (ND) and non-traceable author statement (NAS). The filtered dataset had 15,988 possible annotations for 18,832 genes.

**Table 7. High-throughput gene expression datasets with chemical perturbations.**

| Dataset | sci-Plex | LINCS-Drug |
|---|---|---|
| Source | scRNA-seq | Microarrays |
| Cell lines | A549, K562, MCF7 | ~100 |
| Number of measurements | 648,857 | 677,159 |
| Number of genes | 5087 | 978 |
| Number of perturbations | 180 | 19,990 (8955*) |

*Unique number of perturbations without double-counting stereoisomers.

### Norman et al

We obtained the Norman dataset (Norman et al, 2019) from GEO (accession ID: GSE133344), where each cell was perturbed with 0, 1, or 2 target genes. The data was processed using SCANPY (Wolf et al, 2018), resulting in 109,738 cells and 2279 genes. The processed dataset included 236 unique genetic perturbations affecting 105 target genes, with 11,726 cells remaining unperturbed. Of the 236 genetic perturbations, 230 were successfully mapped to the GO annotation dataset. For benchmarking, we randomly selected three different types of perturbations: (1) one gene perturbed and one gene unobserved, (2) two genes perturbed and one gene unobserved, and (3) two genes perturbed and two genes unobserved. This ensures that all test perturbations contain unobserved gene targets. The data was split with an overall 50–50 ratio for each split.

### Ursu et al

We obtained the Ursu dataset from GEO (accession ID: GSE161824) and filtered the raw data based on the processed datasets. We then concatenated the two datasets containing KRAS and TP53 variants, using the common genes between them. The concatenated data, consisting of 164,931 A549 cells and 1629 genes, was preprocessed using the SCANPY pipeline. Variants were collected based on modifications to the original KRAS and TP53 protein sequences. Since many perturbations were represented by only one or a small number of cells, we filtered out perturbations with fewer than 400 associated cells. This resulted in 80 unique KRAS sequences and 83 TP53 protein sequences with 162,532 cells. For benchmarking, the data was split into training and test sets using an 80–20 ratio, with 32 sequences randomly selected for each test split.

### Jorge et al

This dataset contains perturb-seq data from human hematopoietic stem/progenitor cells (HSPC). We obtained the Jorge dataset from GEO (accession ID: GSE215253), and the accompanying metadata to match cells with their corresponding perturbations was kindly provided by Dr. Jorge D Martin-Rufino. We filtered out cells that could not be matched with any entry in the metadata. The data were preprocessed following the SCANPY pipeline. Next, we mapped the variants at the genome level to the GATA1 sequence, using its canonical version as a reference. Only missense mutations within coding regions were considered valid perturbations. Mutations in non-coding regions, such as the 5'-UTR, were filtered out, as they are identical to wild-type sequences but are still likely to affect gene expression regulations. In addition, mutations that alter the start or stop codons were also filtered out. The final benchmarking dataset consists of 142,872 cells, 2477 genes, and 257 perturbations, including the wild-type sequence. The data were

**Table 8. High-throughput gene expression datasets with CRISPR activation and coding sequence mutations.**

| Dataset | Norman et al | Jorge et al | Ursu et al |
|---|---|---|---|
| Source | scRNA-seq | scRNA-seq | scRNA-seq |
| Cell lines | K562 | HSPCs | A549 |
| Number of measurements | 109,738 | 142,872 | 162,532 |
| Number of genes | 2279 | 2477 (2563*) | 1629 |
| Number of perturbations | 230 | 257 | 163 |
| Perturbation identity | CRISPRa | GATA1 mutations | TP53/KRAS mutations |

*Extra cell markers are added for the novel variants' effect prediction tasks.

split into training and test sets using an 80–20 ratio, with 52 sequences randomly selected for each test split. For the downstream analysis of predicted novel GATA1 mutations, such as the cell-type annotation, we kept cell-type markers in addition to the highly variable genes selected by SCANPY. These markers were collected from the original publication (Martin-Rufino et al, 2023), and the cell-type-specific differentially expressed genes were sourced from the dataset provided by the Dynamo package (Qiu et al, 2022). So the genes in the final dataset are 2563.

The three high-throughput gene expression datasets with CRISPR activation and coding sequence mutations are summarized in Table 8.

## ChemicalVAE

The commonly used one-hot encoding approach can transform drug treatment labels to a vector of 1's and 0's, but it needs pre-specifying the total number of possible drug treatments and cannot encode new treatments after the specification. Therefore, we consider flexible representations $Y$ for drug treatments to predict drug treatment effects on single-cell data for unseen perturbations.

A drug treatment contains abundant information more than just a label such as "S1096". Its pharmacological properties are usually determined by its chemical structure. We thus aim to encode drugs' chemical structures to dense representations. We consider drug treatments' simplified molecular-input line-entry system (SMILES) strings, which distinctively represent chemical structures and treatment information. Although SMILES strings can be encoded to numerical representations through molecular Morgan fingerprints (Rogers et al, 2010) or through language models (Chithrananda et al, 2020; Xu et al, 2017), the representations from these methods are deterministic, meaning that the representations remain the same in replicated encoding implementations. Given that a chemical screen experiment usually contains a limited number of distinct drug treatments, the use of stochastic representations of the drug treatments prevents possible model overfitting.

To improve the learning capacity, especially for representations of unseen treatments, we consider using a chemical variational autoencoder (ChemicalVAE) to generate the stochastic sampled representation $Y$ of each drug's SMILES string (Kusner et al, 2017; Zhu et al, 2021). In essence, the ChemicalVAE first transforms and standardizes SMILES strings to their canonical forms and tokenizes

each canonical SMILES to be encoded as a one-hot matrix. For a canonical SMILES string, the $i$th row of its one-hot matrix corresponds to its $i$th place, and has the $j$th column being 1 and all other columns being 0's, if its $i$th place has the $j$th character in the collected chemical elements' library. The one-hot matrices of SMILES strings are then fitted into ChemicalVAE which provides representations $Y$ for SMILES strings of drug treatments $g$.

We followed the ChemicalVAE model utilized in the paper (Gómez-Bombarelli et al, 2018) and adapted it to PyTorch implementations. The ChemicalVAE model takes each input of size of 120 by 35, and has three one-dimensional convolution layers with the triplet of number of input channels, number of output channels and kernel size being (120, 9, 9), (9, 9, 9) and (9, 10, 11), respectively. There are a Tanh activation function and a batch normalization layer following each convolution layer. After these transformations, the input is then flattened to a fully connected (FC) hidden layer with 196 neurons, and is subsequently activated by a Tanh function, followed by a dropout regularization with a dropout probability of 0.08 and a batch normalization layer. Then two hidden layers both with 196 neurons generate means and standard deviations of the latent variable. The decoder of the ChemicalVAE model has a FC hidden layer with 196 neurons, followed by a Tanh activation, a dropout regularization with a dropout probability of 0.08 and a batch normalization layer. Then the elements of the input are repeated 120 times to be put in a GRU layer with three hidden layers of 488 hidden neurons, followed by a Tanh activation. The input is then transformed to a two-dimensional tensor to be put in a FC layer with 35 neurons and a softmax activation function. Then each input is reshaped to be the output tensor of 120 by 35.

We implemented the ChemicalVAE training on the ZINC data with different learning rates. We finally had an optimal training with a batch size of 128 and a learning rate of $10^{-4}$ for 525 epochs. All related parameters are also summarized in Appendix Table S2.

## GenotypeVAE

For gene knockdowns, most of the existing methods one-hot-encode the target genes across a set of genes (Dixit et al, 2016) or all genes on a coding sequence (Jianzhu Ma et al, 2018). However, this strategy cannot generalize to perturbations with an unseen target gene.

To encode genetic perturbations, we propose a more parsimonious framework and refer to it as GenotypeVAE. Our key insight is that the numerous functional annotations of each gene (organized into a hierarchy in the gene ontology) provide features for learning a low-dimensional representation of individual genes and groups of genes. Using gene ontology (GO) terms, we can represent each target gene $g$ as a one-hot vector $B_g$, where 1's in the vector element correspond to a particular term indicating that the gene has the annotation. Our approach is inspired by Chicco et al (Chicco et al, 2014). If we have a genetic perturbation with multiple target genes $\{g_1, \ldots, g_k\}$, we use annotation-wise union operations to generate a one-hot annotation vector for the genetic perturbation as follows:

$$B_{g_1, \cdots, g_k} = \cup_{j=1}^k B_{g_j}.$$

Then, we can train GenotypeVAE using one-hot representations of many possible genetic perturbations. We use the GO Consortium

gene ontology annotation dataset of human genes. This resource annotates 18,832 genes with 15,988 annotation terms (after removing some annotations with insufficient information). We take the 15,988-dimensional annotation vector as the input to the GenotypeVAE encoder consisting of two hidden layers with 512 and 256 neurons, following output layers for means and standard deviations, both with 10 neurons. The GenotypeVAE decoder also has two hidden layers with 256 and 512 neurons, along with an output layer of 15,988 neurons activated by the sigmoid activation function. We also have a batch normalization layer, Leaky Rectified Linear Unit (ReLU) activation and a dropout layer with a dropout probability of 0.2 following each hidden layer of GenotypeVAE.

We adjusted different learning rates, batch size and epochs. We finally trained GenotypeVAE on the annotation vectors of single and double target genes from the GO annotation dataset with batch size of 128 for 300 epochs at a learning rate of $10^{-4}$. All related parameters are also summarized in Appendix Table S3.

## ESM

A coding variant can be uniquely represented by the protein sequence resulting from the nucleotide alterations induced by CRISPR/Cas9 editing. Similar to chemical perturbations, coding variants can also be summarized as sequences of strings. A key difference is that each character of a protein sequence is a naturally occurring character sequence, whereas a chemical structure is actually a three-dimensional structure (even if it is sometimes represented as a string).

We therefore consider a state-of-the-art language model for protein sequences. Rather than designing our own model and training it from scratch, we employ the previously published Evolutionary Scale Modeling (ESM) (Rives et al, 2021) architecture. ESM is a self-supervised transformer model (Devlin et al, 2018) and was previously shown to achieve better representations and prediction performance on protein sequences compared to other language models such as long short-term memory (LSTM) networks. As with other transformer models (Vaswani et al, 2017), the ESM model was pretrained on large protein sequence datasets (Rao et al, 2021). We adopt a pretrained ESM model specialized for the prediction of single variant effects (ESM-1V, embedding dimension 1280) (Meier et al, 2021), because this application is most similar to our scenario.

However, the representation obtained from ESM is deterministic for a given protein sequence. The fixed protein representations limit the amount of training data available for PerturbNet, especially when there is a small number of protein sequences. We therefore add low-variance noise $\epsilon$ to the ESM representation $Y_{ESM}$ from ESM. The final perturbation representation is thus computed as

$$Y = Y_{ESM} + \epsilon,$$

where $\epsilon \sim \mathcal{N}(0, \sigma^2 I)$. We choose the variance $\sigma^2$ to be a positive constant small enough that it does not significantly alter the relative distances between proteins in the ESM latent space.

## Cell representation network

We used a standard variational autoencoder (VAE) with a Gaussian likelihood as the cell representation network for normalized data.

The objective of the loss function is to maximize the evidence lower bound (ELBO):

$$\mathcal{L} = \mathbb{E}_{q(z|x)}[\log p(x|z)] - \text{KL}\left(q(z|x) \| p(z)\right)$$

In this formula, x represents the input data, which are the gene expressions, and z is the latent variable, where $z \sim \mathcal{N}(0, I)$. We trained the model with a batch size of 128, a learning rate of $10^{-4}$, and for 150 epochs. The latent variable dimension is 10.

For count data, we employed the scVI model from scvi-tools (0.7.1) (Lopez et al, 2018). The scVI model is a VAE that uses discrete likelihoods such as negative binomial, Poisson, and zero-inflated negative binomial (ZINB). We specifically use the scVI model with ZINB likelihood. Since scVI requires the library size to generate new samples, we randomly sample the estimated library size from the training splits during the generation process. We trained the scVI model with default settings for 700 epochs. The latent variable dimension is 10.

All related parameters are also summarized in Appendix Table S4.

## Conditional invertible neural network (cINN)

We consider employing complex normalizing flows of invertible neural networks to understand the relationship between perturbation representation and cellular responses. An affine coupling block (Dinh et al, 2016) enables the input $U = (U_1^T, U_2^T)^T$ to be transformed to output $W = (W_1^T, W_2^T)^T$ with:

$$W_1 = U_1 \odot \exp\{\text{scale}_1(U_2)\} + \text{trans}_1(U_2)$$

and

$$W_2 = U_2 \odot \exp\{\text{scale}_2(W_1)\} + \text{trans}_2(W_1),$$

where $\text{scale}_1(\cdot)$, $\text{scale}_2(\cdot)$, $\text{trans}_1(\cdot)$, $\text{trans}_2(\cdot)$ are arbitrary scale and transformation neural networks, and $\odot$ is the Hadamard product or element-wise product. The inverse of the coupling blocking can be represented by

$$U_2 = \{W_2 - \text{trans}_2(W_1)\} \oslash \exp\{\text{scale}_2(W_1)\}$$

and

$$U_1 = \{W_2 - \text{trans}_1(U_2)\} \oslash \exp\{\text{scale}_1(U_2)\},$$

where $\oslash$ is the element-wise division. The affine coupling block allows bijective transformations between $U$ and $W$ with strictly upper or lower triangular Jacobian matrices. A conditional coupling block is further adapted to concatenate a conditioning variable with inputs in scale and transformation networks. A conditional coupling block preserves the invertibility of the block and the simplicity of the Jacobian determinant.

A conditional invertible neural network (cINN) (Ardizzone et al, 2019; Rombach et al, 2020) is a type of conditional normalizing flow with conditional coupling blocks and activation normalization (actnorm) layers (Kingma et al, 2018), with both forward and inverse translations. Denote representations from two domains as $Y \in \mathcal{D}_Y$ and $Z \in \mathcal{D}_Z$. A cINN modeling $Z$ over $Y$ gives forward translation

$$Z = f(V|Y)$$

and inverse translation

$$V = f^{-1}(Z|Y),$$

where $V \sim \mathcal{N}(0, I)$. The cINN effectively models $p(Z|Y)$, the probabilistic dependency of $Z$ over $Y$ with a residual variable $V$. As a cINN seeks to extract the shared information from $Y$ and add residual information $V$ to generate $Z$, the objective function to train a cINN is the Kullback–Leibler (KL) divergence between the residual's posterior $q(V|Y)$ and its prior $p(V)$. The objective function can further be derived to

$$\mathbb{E}_{p(Y)}[D_{\text{KL}}\{q(V|Y)\|p(V)\}] = \mathbb{E}_{p(Z,Y)}\left[-\log p\{f^{-1}(V|Y)\} - \left|\det J_{f^{-1}}(Z|Y)\right|\right] - H(Z|Y),$$

(1)

where $\det J_{f^{-1}}$ is the determinant of the Jacobian matrix of $f^{-1}$ and $H$ is a constant entropy. The optimal $f$ that minimizes the objective function in Eq. (1) gives $q(V|Y) = p(V)$. In addition, the objective is an upper bound of the mutual information $I(V, Y)$. Therefore, a well-trained cINN effectively achieves independence between $V$ and $Y$. cINN has the same parameters for forward and inverse translations, reducing the number of model parameters while still preserving network details in both translation directions, and has been utilized to translate domain representations of images and texts (Rombach et al, 2020).

We trained the cINN translations following Rombach et al (Rombach et al, 2020), where a cINN consists of 20 invertible neural network blocks and an embedding module. Each block has an alternating affine coupling layer, an actnorm layer and a fixed permutation layer. The embedding module consists of FC hidden layers and Leaky ReLU activation functions to embed the conditioning variable into a 10-dimensional variable. We fixed the batch size of 128, the learning rate of $4.5 \times 10^{-6}$ and varied different numbers of epochs for training cINN. We trained the cINN for 50 epochs on the Norman et al, Ursu et al, and Jorge et al datasets, while for the LINCS-Drug and sci-Plex datasets, the cINN required 100 epochs to converge. All related parameters are also summarized in Appendix Table S5.

## PerturbNet generative process

The generative process used by PerturbNet to make predictions is outlined below:

**Algorithm 1**: Generative Process of PerturbNet
**Input**: Drug treatment of interest g.
1. Generate the Perturbation representation $Y$ using perturbation represenation network
2. Sample noise from the prior of cINN, generate the conditional cell representations $Z|Y$
3. Cell representation network decodes the $Z|Y$ to generate predicted single cells $\mathcal{X}'$. For scVI, sample library sizes $l_n$ from library size of the training data $\{l_n^{train}\}$ before decoding.

**Result**: predicted single cells $\mathcal{X}'$ under perturbation g.

## Baseline mean and median model

The mean and median models predict gene expression using the mean and median values of each gene in the training dataset.

## Baseline training sample model

We propose a naive training sample model in Algorithm 2, which randomly samples single-cell data from treatments within the training split. If the perturbation has a very subtle effect, these random samples from the observed data can serve as reasonable approximations, particularly when evaluating larger gene sets. However, when focusing on a subset of genes specifically affected by the perturbation, such as differentially expressed genes (DEGs) and large-effect genes, a reduction in the model's performance is expected.

**Algorithm 2**: Baseline Random Model
**Input**: Drug treatment of interest g. A set of single-cell samples in train split $\{\mathcal{X}_{train}\}$ receiving drug treatments other than g.
1. Sample a number of cells $\mathcal{X}'$ with replacement from $\mathcal{X}_{train}$.
**Result**: predicted single cells $\mathcal{X}'$ under perturbation g.

## Baseline KNN model

From the perturbation representations, $Y$, of drug treatments, we can learn the relationship of several drug treatments in their latent space. We assume that drug treatments with close latent values tend to also have similar single-cell responses. Thus, the distributions of perturbation responses $p(X|G = g_1)$ and $p(X|G = g_2)$ are similar if $g_1$ and $g_2$ have close representations of $y_1$ and $y_2$.

We then propose our baseline model using the $k$-nearest neighbors (KNN) algorithm to predict single-cell data under drug treatments in Algorithm 3. From ChemicalVAE, we can obtain the representation $Y$ for a set of treatments $\mathcal{G}$, each of which has measured single-cell samples. Then for a drug treatment $g \notin \mathcal{G}$ with representation $y$, we can find its $k$-nearest neighbors $\{g_{(1)}, \ldots, g_{(k)}\}$ from $\mathcal{G}$ based on $Y$. We then sample single-cell samples treated with the $k$-nearest treatments in proportion to their exponentiated negative distances to the treatment of interest in the latent space of $Y$. The sampled single-cell data can be regarded as a baseline prediction for the single-cell data with the treatment of interest.

**Algorithm 3**: Baseline KNN Model
**Input**: Drug treatment of interest g and its representation $y$. A set of drug treatments $\mathcal{G} = \{g_1, \ldots, g_m\}$ with their representations $\{y_1, \ldots, y_m\}$ as well as single-cell sample sets $\{\mathcal{X}_1, \ldots, \mathcal{X}_m\}$.
1. Train KNN algorithm (k = 5) on $\{y_1, \ldots, y_m\}$.
2. Obtain $y$'s k neighbors $\{y_{(1)}, \ldots, y_{(k)}\}$ and their pairwise distances $(d_{(1)}, \ldots, d_{(k)})^T$ from the trained KNN algorithm.
3. Sample a number of cells $\mathcal{X}'$ through stratified sampling with replacement from $\{\mathcal{X}_{(1)}, \ldots, \mathcal{X}_{(k)}\}$. Each set $\mathcal{X}_{(i)}$ has a proportion of $\exp\{-d_{(i)}\}/\sum_{j=1}^{k} \exp\{-d_{(j)}\}$.
**Result**: predicted single cells $\mathcal{X}'$ under perturbation g.

## Baseline linear model

For chemical perturbations and genetic perturbations that introduce coding variants, we used a naive linear regression model as the linear baseline. The model is specified as follows:

$$\arg\min_{W} \| \mathbf{X}^{\text{train}} - (\mathbf{P}^T\mathbf{W} + \mathbf{b}) \|_2^2$$

Where $\mathbf{X}^{\text{train}}$ is the gene expression matrix for each observation ($n_{\text{obs}} \times n_g$), $\mathbf{W}$ is a weight matrix of size $n_p \times n_g$, and $\mathbf{P}$ is a perturbation representation matrix of size $n_p \times n_{\text{obs}}$, generated by the corresponding perturbation models. For chemical perturbations, $\mathbf{P}$ is generated by ChemicalVAE, and for coding variants, $\mathbf{P}$ is generated by the ESM model with added Gaussian noise. For numerical stability, we used a ridge penalty equal to 0.1.

For CRIPSRa or CRISPRi predictions, we benchmarked against a linear baseline model that outperforms current deep learning methods (Ahlmann-Eltze et al, 2024). This linear baseline assumes that post-perturbation gene expressions can be linearly predicted using gene and perturbation representations:

$$\arg\min_{W} \| \mathbf{X}^{\text{train}} - (\mathbf{GWP}^T + \mathbf{b}) \|_2^2$$

where $\mathbf{X}^{\text{train}}$ is the pseudobulked per condition of the single-cell gene expressions. Here, we aggregated the data by taking the mean of the expressions. Each row of $\mathbf{X}^{\text{train}}$ corresponds to a readout gene, and each column corresponds to a perturbation, with the values representing the mean gene expressions induced by the respective perturbations. The dimension of $\mathbf{X}^{\text{train}}$ is $n_g \times n_p$, where $n_g$ is the number of genes and $n_p$ is the number of single-gene perturbations in the training set. $\mathbf{W}$ is a $k \times k$ weight matrix. $\mathbf{G}$ is the gene representation generated by applying PCA on $\mathbf{X}^{\text{train}}$. The dimension of $\mathbf{G}$ is $n_g \times k$, where k is the number of PCs. $\mathbf{P}$ is the perturbation representation, which is a subset of $\mathbf{G}$ consisting of rows corresponding to the genes that were perturbed in the training data, giving $\mathbf{P}$ a dimension of $n_p \times k$. $\mathbf{b}$ is the vector of row means of the training data. The model solves for the weight matrix $\mathbf{W}$ using a ridge penalty of $\lambda = 0.1$ for numerical stability:

$$\mathbf{W} = (\mathbf{G}^T\mathbf{G} + \lambda\mathbf{I})^{-1}\mathbf{G}^T(\mathbf{X}^{\text{train}} - \mathbf{b})\mathbf{P}(\mathbf{P}^T\mathbf{P} + \lambda\mathbf{I})^{-1}$$

Then the model predicts gene expression induced by the single-gene perturbations in test splits by:

$$\hat{\mathbf{X}} = \mathbf{b} + \mathbf{GWP}_{\text{test}}^T$$

For two-gene perturbations, we simply add the predictions of single-gene perturbations from the linear model. Thus, for perturbing gene A and gene B, the predicted expression is:

$$\hat{\mathbf{X}}_{\mathbf{AB}} = \hat{\mathbf{X}}_{\mathbf{A}} + \hat{\mathbf{X}}_{\mathbf{B}} - \mathbf{X}_{\text{control}}$$

We do not use the additive model from the original paper because it requires real gene expressions from single-gene perturbations, meaning that such a model cannot predict two-gene perturbations when either gene has not been previously observed.

## Prediction metrics

We use the $R^2$ and Pearson correlation metrics to evaluate the prediction performance of different models. Our analysis considers

three distinct gene sets: highly variable genes, the top 50 differentially expressed gene (DEGs), and large-effect genes. Highly variable genes represent an informative subset of all measured genes and were identified using SCANPY. Given that most perturbations exhibit limited effects, typically inducing notable changes only in a subset of highly variable genes, we also calculated metrics based on the top 50 DEGs. The DEGs were identified using SCANPY's *t* test (code is shown below), as we have a large number of cells subjected to each perturbation. The large-effect genes are a subset of the top 50 DEGs, consisting of genes with absolute log-fold changes greater than 1. This subset emphasizes genes significantly affected by the perturbation, providing insight into model performance in cases where the perturbation effect is subtle.

```
sc.tl.rank_genes_groups(adata,
n_genes = 50,
method = "t-test",
corr_method = "benjamini-hochberg",
groupby = groups,
reference = control)
```

### $R^2$

We follow the use of the $R^2$ metric, which has been employed in several related frameworks, to evaluate single-cell responses to perturbations (Hetzel et al, 2022; Lotfollahi et al, 2020, 2023, 2019). To ensure a fair comparison, we calculate $R^2$ on the normalized data for both predicted and real data. When the data is already normalized (as in the LINCS-Drug dataset), we calculate $R^2$ directly. For discrete data, we first normalize the total number of counts per cell to $10^4$, followed by log-transformation. We then compute the mean gene expression values for every gene in the selected gene set of both predicted and real cells. By definition, $R^2$ represents the proportion of variance explained by the model, with higher $R^2$ values indicating a better model fit. The $R^2$ metric ranges from $(-\infty, 1]$. The formula of $R^2$ is shown below, where $y_i$ denotes the observed mean gene expression, $\hat{y}_i$ is the predicted mean gene expression, and $n$ is the number of genes.

$$R^2 = 1 - \frac{\sum_{i=1}^n (y_i - \hat{y}_i)^2}{\sum_{i=1}^n (y_i - \bar{y})^2}$$

#### *Pearson correlation*

We apply the same preprocessing steps to both the predicted and real data as we did when calculating $R^2$. Pearson's correlation measures the linear relationship between the predictions and the real data. A higher Pearson correlation generally indicates that the model provides a better fit to the data. The range of Pearson correlation values is $[-1, 1]$. The formula of Pearson correlation is shown below, where $y_i$ denotes the observed mean gene expression, $\hat{y}_i$ is the predicted mean gene expression, and $n$ is the number of genes.

$$r = \frac{\sum_{i=1}^n (y_i - \bar{y})(\hat{y}_i - \bar{\hat{y}})}{\sqrt{\sum_{i=1}^n (y_i - \bar{y})^2}\sqrt{\sum_{i=1}^n (\hat{y}_i - \bar{\hat{y}})^2}}$$

### Differentially expressed genes selection

We used SCANPY to extract the top 50 differentially expressed genes for each dataset. Since there were sufficient observations

within each perturbation group, we applied a t test, using the control group as the reference.

### Analysis of non-additive interactions effect on PerturbNet's performance

We use the genetic interaction annotations directly from Norman et al (Norman et al, 2019) Since the accuracy of predictions is largely influenced by the number of unseen genes when comparing PerturbNet to other methods (Figs. 3E,F and EV3), we applied linear regression on performance metrics ($R^2$) to analyze how non-additive genetic interactions affect PerturbNet's predictions. We included several potential confounding factors: the number of cells in each perturbation group in the dataset, the number of GO annotations (as a complexity measure of the predicted perturbation), perturbation type (e.g., "0/2" represents two perturbed genes, both unseen in the training set), the minimum latent space distance between the predicted and training set perturbations, the number of large-effect genes caused by the predicted perturbation (as a magnitude measure of the perturbation effect), and the types of genetic interactions (with no interactions or unknown interactions as the reference level). These models were fitted on $R^2$ calculated for all highly variable genes and for differentially expressed genes (DEGs). Detailed results are provided in Appendix Tables S6 and S7.

### Analysis of the predicted GATA1 mutations

The predicted count data was processed using SCANPY before downstream analysis. We utilized UMAP to visualize the predicted cell distributions and projected the mean expression levels, representing each mutation, onto the same UMAP space. The Leiden algorithm was applied for cell clustering, and the clusters were manually annotated based on the expression of cell-type-specific markers. These markers were collected from the original publication (Martin-Rufino et al, 2023) or identified as differentially expressed genes using the dataset provided by the Dynamo package (Qiu et al, 2022). UMAP of the marker representatives are displayed in Appendix Fig. S6. Mutation clusters were also identified using the k-means algorithm on the UMAP. Based on the density of erythroid cells, we classified three mutation clusters as erythroid-depleted, erythroid-intermediate, and erythroid-enriched.

Logistic regression analysis was used to explore potential factors associated with cluster assignment probability (Appendix Tables S8–S10). Since mutations in different clusters have distinct effects, the significant factors tied to cluster assignment are the ones influencing these mutation effects. We performed three logistic regressions for each pair of cluster assignments, with potential predictors including volume difference, KD value difference, and mutation position categories. Volume difference refers to the changes between mutated GATA1 and wild-type GATA1, with a similar definition for KD value difference. The KD value is measured by the Kyte-Doolittle scale, which assesses the hydrophobicity of amino acids (Kyte et al, 1982). The collected amino acid properties are listed in (Appendix Table S11).

Mutation positions were categorized as "Other residues," "Close to DNA-contact residue," "DNA-contact," and "Wild type" (indicating the sequence without mutations). These categories were determined based on the distance between the $\alpha$ - **C** atom of each

residue and the DNA in the GATA1–DNA complex structure (PDB ID: 3VD6). Any residue within 4 Å of the DNA was classified as a "DNA-contact" residue, and neighboring residues were categorized as "Close to DNA-contact residues."

Next, we calculated log-fold changes for large-effect genes in each protein sequence. For each position, we selected the mutation with the largest effect, then ranked this subset based on mean log-fold changes. Mutations at residues categorized as "DNA-contact" or "Close to DNA-contact residues" were considered critical. The structure of the GATA1–DNA complex was visualized using PyMOL (DeLano, 2002).

### Integrated gradients

As we connect perturbation and cell state in PerturbNet, we can interpret how a perturbation changes the cell state distribution by predicting cellular representations using PerturbNet. We can further interpret the effects of features and components of the perturbation with the state-of-the-art AI methods. Denote $F(\cdot)$ as a function taking input feature vector $\boldsymbol{T} = (T_1, \ldots, T_n)^T \in \mathbb{R}^n$ to generate output in [0, 1]. Then its attribution is a vector $\boldsymbol{A} = (a_1, \ldots, a_n)^T$ and each value $a_i$ is the contribution of $T_i$ to the prediction of $F(\boldsymbol{T})$.

Previous attempts to interpret neural network models have focused on gradients (Baehrens et al, 2010; Simonyan et al, 2013) and back-propagation (Shrikumar et al, 2017). We use the method of integrated gradients (Sundararajan et al, 2017), which has been applied to interpret deep learning models across a range of domains, including computational chemistry (McCloskey et al, 2019). The attribution score of the integrated gradients method for the $i$th dimension of input $\boldsymbol{T}$ is defined as

$$a_i = (T_i - T_{0,i}) \int_{\alpha=0}^{1} \frac{\partial F\{\boldsymbol{T}_0 + \alpha(\boldsymbol{T} - \boldsymbol{T}_0)\}}{\partial T_i} d\alpha,$$

where $\boldsymbol{T}_0 = (T_{0,0}, \ldots, T_{0,n})^T$ is a baseline input.

A prediction neural network model on cellular representation can be formulated from PerturbNet as $\boldsymbol{Z} = f(\boldsymbol{V}|\boldsymbol{Y})$ and $\boldsymbol{Y} = h(\boldsymbol{B})$ ($\boldsymbol{B}$ is the input perturbation feature matrix, such as the one-hot matrix). The input $\boldsymbol{T}$ can be formulated as $(\boldsymbol{V}^T, \boldsymbol{Y}^T)^T$ or $(\boldsymbol{V}^T, \boldsymbol{B}^T)^T$. In addition, a classification neural network model on $\boldsymbol{Z}$ provides a classification score within [0, 1]. We can then find input features that increase the probability of generating cells in a particular cell state.

## Data availability

All datasets analyzed here are previously published and freely available. Processed datasets and computer code produced in this study are available in the following databases: processed dataset and model weights: Hugging Face (https://huggingface.co/cyclopeta/PerturbNet_reproduce); PerturbNet code and tutorials: GitHub (https://github.com/welch-lab/PerturbNet).

The source data of this paper are collected in the following database record: biostudies:S-SCDT-10_1038-S44320-025-00131-3.

## Peer review information

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

## Acknowledgements

The authors would like to thank Hojae Lee, Yuwei Bao, Andrew Robbins, Ruoxi Gao, Albert Hung, and Jialin Liu for helpful discussions. The authors thank Jorge D. Martin-Rufino for sharing mutation metadata for their published dataset. The authors thank Quancheng Liu, Xiheng Ren, and Zhiran Wang for exploring applications of this method. This work was supported by NIH grants R01HG010883 and U01HG011952 to JDW.

## Author contributions

**Hengshi Yu**: Conceptualization, Data curation, Software, Investigation, Visualization, Methodology, Writing—original draft, Writing—review and editing. **Weizhou Qian**: Conceptualization, Data curation, Software, Visualization, Methodology, Writing—original draft, Writing—review and editing. **Yuxuan Song**: Formal analysis, Visualization, Writing—review and editing. **Joshua D Welch**: Conceptualization, Data curation, Supervision, Funding acquisition, Visualization, Methodology, Writing—original draft, Project administration, Writing—review and editing.

Source data underlying figure panels in this paper may have individual authorship assigned. Where available, figure panel/source data authorship is listed in the following database record: biostudies:S-SCDT-10_1038-S44320-025-00131-3.

## Disclosure and competing interests statement

The authors declare no competing interests.

# Expanded View Figures

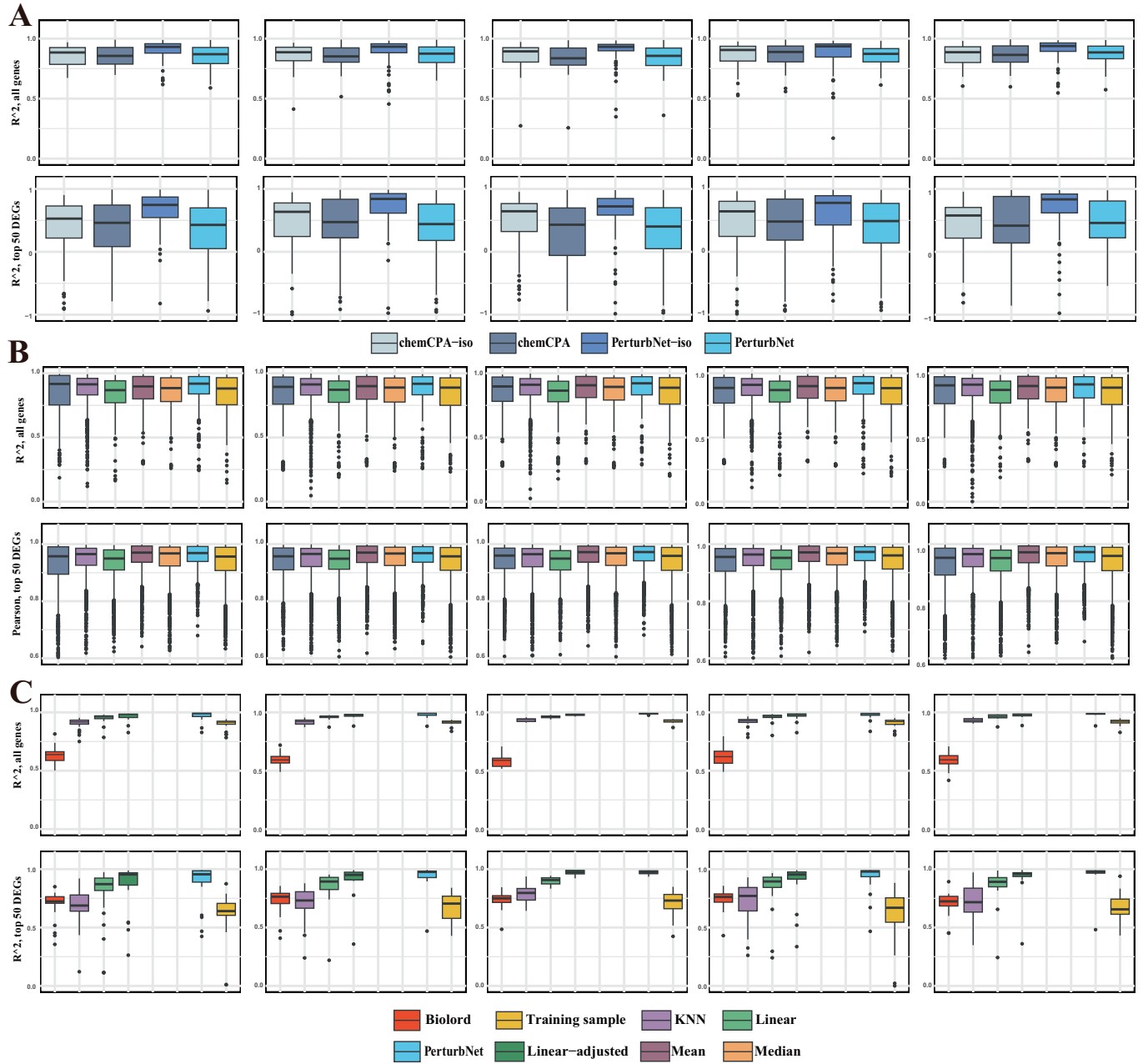

**Figure EV1. Box plots of evaluation metrics on the LINCS-Drug and sci-Plex datasets.**

(A) Comparison between PerturbNet and chemCPA trained with and without stereoisomers. (B) Benchmark results on the LINCS-Drug dataset. (C) Benchmark results on the sci-Plex dataset. Note: for all box plots in this panel, the box plots show the median (center line), the 25th and 75th percentiles (box bounds), and 1.5× the interquartile range (whiskers). Points beyond the whiskers are plotted as outliers.

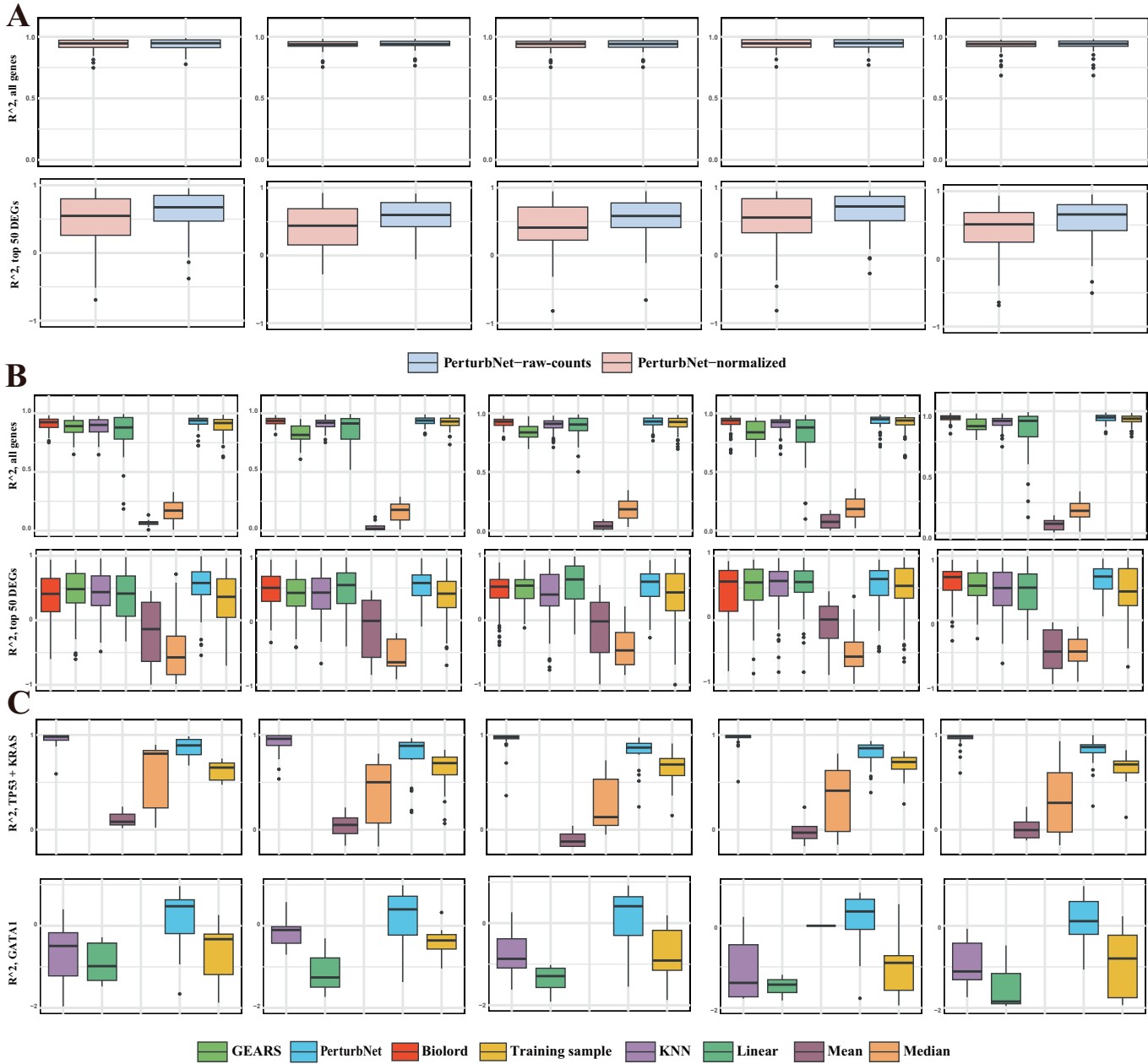

**Figure EV2.  Box plots of evaluation metrics on the genetic perturbation datasets.**

(A) Comparison between PerturbNet trained on raw counts and on normalized expression. (B) Benchmark results on the Norman et al dataset. (C) Benchmark results on the Ursu et al and Jorge et al datasets. Note: for all box plots in this panel, the box plots show the median (center line), the 25th and 75th percentiles (box bounds), and 1.5× the interquartile range (whiskers). Points beyond the whiskers are plotted as outliers.

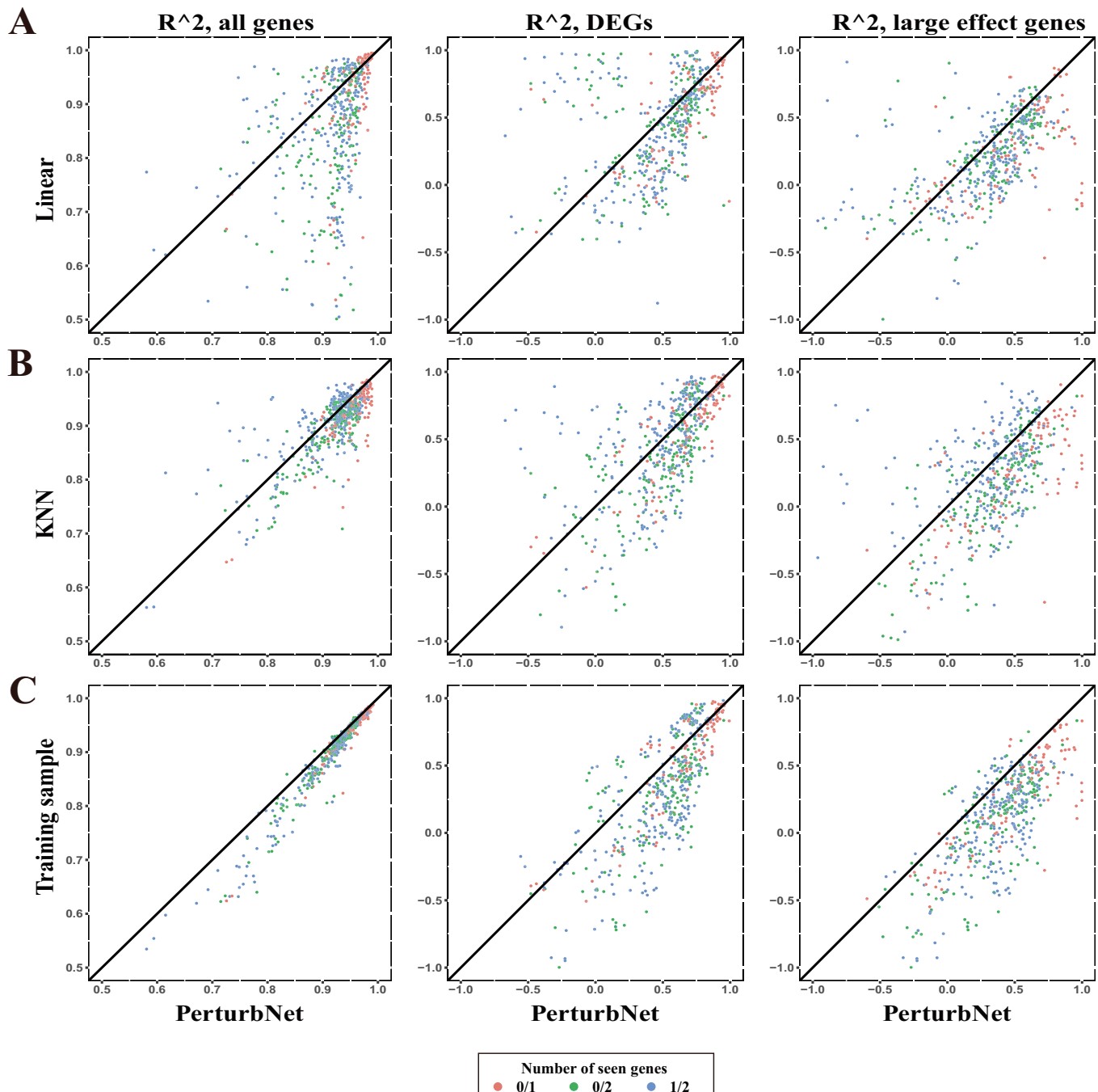

**Figure EV3. Scatter plots of $R^2$ values for unseen genetic perturbations, calculated across all genes, the top 50 differentially expressed genes (DEGs), and large-effect genes from the Norman et al dataset.**

Data points are aggregated from all five test splits. Different colors represent the "number of seen genes." Labels such as "0/1" indicate that the test perturbation affects one unseen gene, while "0/2" indicates two unseen genes are perturbed. "1/2" denotes that two genes are perturbed, but one of the target effects has already been observed independently or in combination with other genetic perturbations. (**A**) PerturbNet v.s linear baseline. (**B**) PerturbNet v.s KNN. (**C**) PerturbNet v.s. training sample.

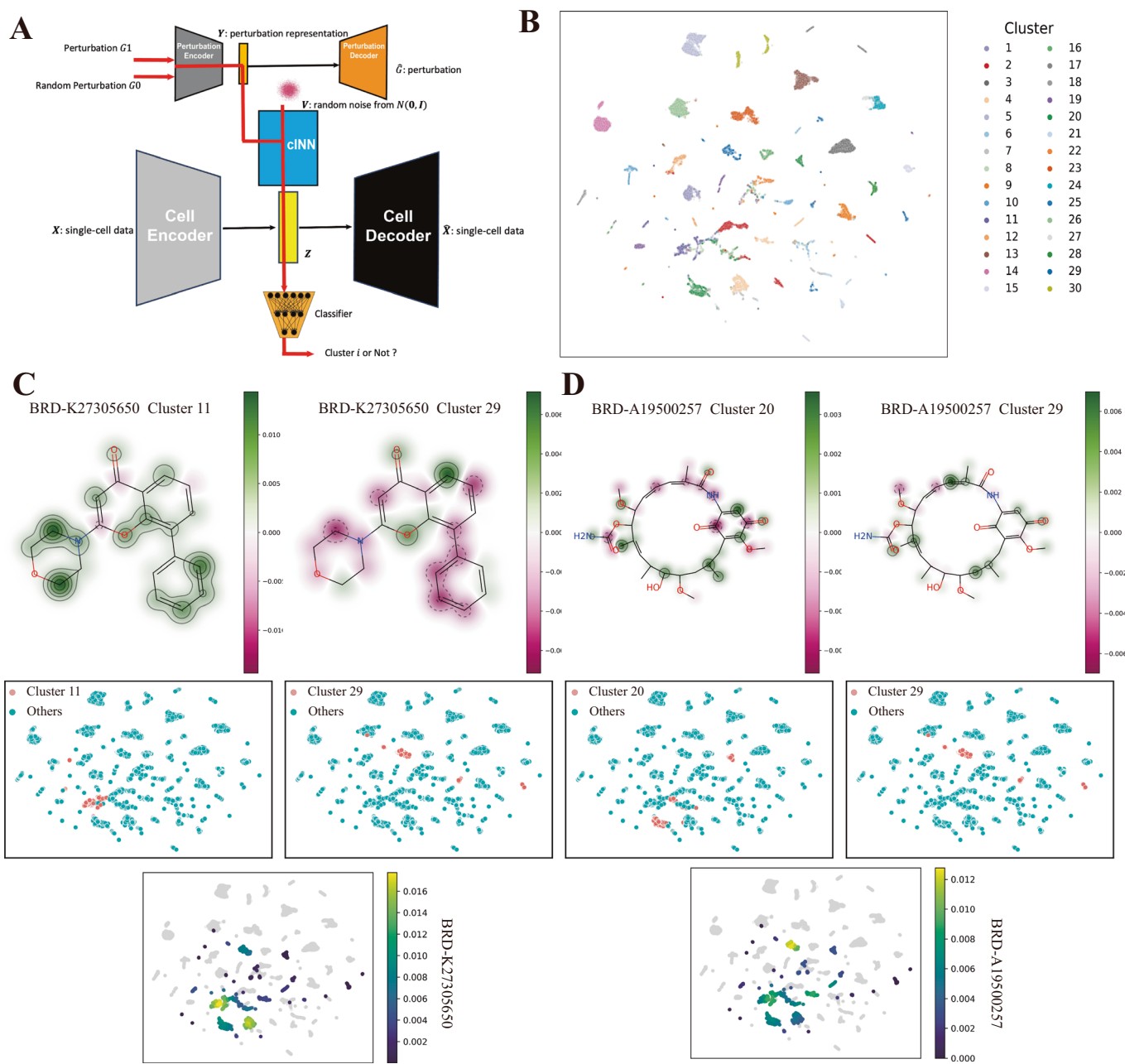

**Figure EV4. Attributing cell state shifts to specific features of perturbations.**

(A) Diagram of approach for attributing perturbation outcomes to specific perturbation features. We attach a binary classifier after the cINN to classify cells into discrete types. By comparing the classification results of an input perturbation and a baseline (random perturbations), we can determine which perturbation features increase classification probability. (B) UMAP plots of cells from LINCS-Drug colored by cluster label. (C, D) UMAP plots of LINCS-Drug with selected clusters and the selected drug colored by attribution scores for each atom. The bottom is the UMAP plot colored by the cell density from selected drugs.

