## [Peer Review File · Molecular Systems Biology]

PerturbNet predicts single-cell responses to unseen chemical and genetic perturbations

Hengshi Yu, Weizhou Qian, Yuxuan Song, and Joshua Welch

Corresponding author(s): Joshua Welch (welchjd@umich.edu)

Review Timeline:

Submission Date:	6th May 25
Editorial Decision:	21st May 25
Revision Received:	3rd Jun 25
Accepted:	17th Jun 25

Editor: Poonam Bheda

Transaction Report:

Reviewer 1

Thank you for the response to my initial review and your efforts to revise the manuscript - in particular the fact that isomers and covariates are now taken into account. However, despite the substantial time since the first submission, the current version still has major structural, methodological, and presentation issues:

- Lack of sufficient effort to address previous comments: The authors have made only minimal revisions to the manuscript, and rather than focusing on improving the main issues with the core of the method, the presentation and validation of its results, chose to expand the application to other domains (protein) without providing robust evidence to give confidence on the new results. The manuscript still lacks the necessary contextualization and supporting data to fully illustrate its advantages and why it may be better than competing baselines. Given the extended time since the initial submission, a more substantial effort was expected.

- Poor presentation: The new figures, in particular the boxplots with R^2 values contain excessive whitespace and compressing the values to a narrow range, making them difficult to interpret. The presentation of different data splits does not add much value (can be left in supplement), but that space could have been better use to make one much clearer plot across splits. The compression of data visualization in the plots prevents an adequate assessment of performance differences between methods, leaving all the interpretation to the stars indicating p-value significance, which as I commented earlier, can be dependent on various parameters and are not a robust way to evaluate the performance competitively. Statement of the numeric performance values for the compared methods are missing (e.g. mentioned in text or in tables), making it impossible to objectively evaluate the effectiveness of the proposed approach. The bottom rows of panels in Figure 2d and 2e present a y-axis that appears meaningless, lacking clear labels and interpretation. The authors have to make a considerably better effort to illustrate the performance of the methods in a objective and quantitative way, and figures should be improved for clarity.

We have updated all main figures to display only a single train/test split per comparison, removed the p-value indicators, and reformatted the plots to allow for better visual comparison of the boxplots. We moved the box plots for the remaining train/test splits to the supplement. To summarize the results across five train/test splits, we included a heatmap illustrating the proportion of cases in which PerturbNet outperformed other methods. Additionally, we provide tables presenting the mean and median values of the evaluation metrics for each dataset to facilitate a clearer comparison between models.

Fig. 2 PerturbNet predicts response to small molecule treatment. (a) Diagram of the chemical variational autoencoder (ChemicalVAE) architecture for encoding small molecules represented as SMILES strings. The network was trained on the ZINC dataset, consisting of approximately 250,000 drug-like molecules. No gene expression information is used for training the drug representation network. Note that a cell representation network is also trained on gene expression values (not shown here). (b) Visualization of PerturbNet predictions for two distinct perturbations from the LINCS-Drug dataset. The UMAP coordinates are computed from the latent spaces of the perturbation network (left) and the cell state network (right). The mapping function learned by the cINN is indicated with lines connecting the perturbation and cell state representations. The predicted cell state distributions are also indicated with contour lines. (c) Box plots of R^2 calculated on all genes and the top 50 differentially expressed genes (DEGs) in one test split of the LINCS-Drug dataset for chemCPA and PerturbNet with (-iso) and without stereoisomers shared between train and test sets. (d) Heatmap showing the proportion of train/test splits in which PerturbNet or ChemCPA (trained with stereoisomers) outperforms the same model trained without stereoisomers across five test splits of the LINCS-Drug dataset. "Outperforms" here means a significant difference by a one-sided Wilcoxon test. (e) Box plots of evaluation metrics in one test split of the LINCS-Drug dataset (plots for additional splits are shown in the supplement). (f) Box plots of evaluation metrics in one test split of the sci-Plex dataset (plots for additional splits are shown in the supplement). Some models are not displayed due to out-of-scale values for certain metrics. (g) Heatmap showing the proportion of train/test splits in which PerturbNet achieves significantly better performance (based

on one-sided Wilcoxon tests) compared to competing models (columns) across five train/test splits of the sci-Plex and LINCS-Drug datasets.

Model	Median R^2	Mean R^2	Median R^2 (DEG)	Mean R^2 (DEG)
Chemcpa-iso	0.891	0.319	0.563	0.319
Chemcpa-iso-free	0.857	0.192	0.418	0.192
PerturbNet-iso	0.933	0.623	0.794	0.623
PerturbNet-iso-free	0.874	0.154	0.394	0.154

Table 1: Summary of mean and median evaluation metrics on the LINCS-Drug dataset for chemCPA and PerturbNet with (-iso) and without stereoisomers shared between train and test sets. The top-performing values are bolded and the second-best are underlined for each metric.

Model	Median R^2	Mean R^2	Median Pearson (DEG)	Mean Pearson (DEG)
Chemcpa	0.899	0.862	0.956	0.931
KNN	0.912	0.873	0.965	0.940
Linear	0.869	0.847	0.950	0.932
Mean	0.901	0.881	0.971	0.954
Median	0.888	0.866	0.968	0.944
PerturbNet	0.919	0.894	0.971	0.958
Training sample	0.887	0.853	0.958	0.936

Table 2: Summary of mean and median evaluation metrics on the LINCS-Drug dataset. The top-performing values are bolded and the second-best are underlined for each metric.

Model	Median R^2	Mean R^2	Median R^2 (DEG)	Mean R^2 (DEG)
Biolord	0.587	0.554	0.716	0.658
KNN	0.918	0.893	0.664	0.328
Linear	0.954	0.936	0.825	0.569
Linear-adjusted	0.976	0.964	0.917	0.824
Mean	-6.271	-6.156	-10.161	-11.838
Median	-0.636	-0.647	-2.296	-2.364
PerturbNet	0.984	0.968	0.951	0.865
Training sample	0.915	0.894	0.596	0.356

Table 3: Summary of mean and median evaluation metrics on the sci-Plex dataset. The top-performing values are bolded and the second-best are underlined for each metric.

- Inconsistent performance: The performance of PerturbNet in unseen perturbations is not consistently better than other baselines. This is particularly acute when not all genes are considered, but a set of likely more relevant (and higher expressed) genes such as differentially expressed or with large effect (Fig. 5d, Figure S4, Figure 3c DEGs). A linear model baseline for Fig. 2 is not shown. This demonstrates that baselines such as linear models or KNNs can do comparatively well.

Following the reviewer's comment, we have added a linear baseline model (both with and without incorporating covariates) to Fig. 2, and PerturbNet significantly outperforms this model. We further added new mean and median baselines following reviewer 2's comments.

We also think that our results are more impressive than what the reviewer suggests. Here is a summary of how the baseline models fare:

- *PerturbNet is statistically significantly better than every baseline model on the SciPlex dataset (Fig. 2). The only exception is that the covariate-adjusted linear model is comparable (not better) on 1 out of 5 train/test splits. If the metrics are averaged across all 5 splits, PerturbNet achieves the best performance.*
- *PerturbNet is statistically significantly better than every baseline model on the LINCS-Drug dataset (Fig. 2). The only exception is that the mean model achieves comparable (not better) performance on 2 out of 5 train/test splits. If the metrics are averaged across all 5 splits, PerturbNet achieves the best performance.*
- *PerturbNet is statistically significantly better than every baseline model on the Norman dataset (Fig. 3). The only exception is that the KNN model achieves comparable (not better) performance on 1 out of 5 train/test splits and the linear model achieves comparable (not better) performance on 2 out of 5 train/test splits. If the metrics are averaged across all 5 splits, PerturbNet achieves the best performance.*
- *The KNN model outperforms PerturbNet on the Ursu dataset in 4 out of 5 train/test splits (Fig. 4). PerturbNet outperforms all baselines on all splits in the Jorge dataset (Fig. 4).*

We note that PerturbNet outperforms current state-of-the-art methods across all of the train-test splits mentioned above (and there are no previous methods that can make predictions for Ursu and Jorge).

We would also like to point out that previous studies have reported only single numbers rather than individual results from multiple train/test splits. The only reason the reviewer can raise this point is because we have been fully rigorous in reporting the results from every train/test split.

We have also updated the visualizations for Fig. 2 , Fig. 3 and Fig. 4 (originally Fig.5) and added tables to facilitate easier comparisons.

Model	Median R ²	Mean R ²	Median Pearson (DEG)	Mean Pearson (DEG)
Chemcpa	0.899	0.862	0.956	0.931
KNN	0.912	0.873	0.965	0.940
Linear	0.869	0.847	0.950	0.932
Mean	0.901	0.881	0.971	0.954
Median	0.888	0.866	0.968	0.944
PerturbNet	0.919	0.894	0.971	0.958
Training sample	0.887	0.853	0.958	0.936

Table. 2: Summary of mean and median evaluation metrics on the LINCS-Drug dataset. The top-performing values are bolded and the second-best are underlined for each metric.

Model	Median R^2	Mean R^2	Median R^2 (DEG)	Mean R^2 (DEG)
Biolord	0.587	0.554	0.716	0.658
KNN	0.918	0.893	0.664	0.328
Linear	0.954	0.936	0.825	0.569
Linear-adjusted	0.976	0.964	0.917	0.824
Mean	-6.271	-6.156	-10.161	-11.838
Median	-0.636	-0.647	-2.296	-2.364
PerturbNet	0.984	0.968	0.951	0.865
Training sample	0.915	0.894	0.596	0.356

Table 3: Summary of mean and median evaluation metrics on the sci-Plex dataset. The top-performing values are bolded and the second-best are underlined for each metric.

Model	Median R^2	Mean R^2	Median R^2 (DEG)	Mean R^2 (DEG)
Biolord	0.936	0.923	0.540	0.436
GEARS	0.858	0.853	0.513	0.451
KNN	0.918	0.901	0.452	0.409
Mean	-0.097	-0.115	-1.447	-1.549
Median	0.170	0.152	-1.408	-1.439
PerturbNet	0.942	0.928	0.629	0.535
Training sample	0.930	0.912	0.430	0.375
Linear	0.891	0.830	0.561	0.489

Table 4: Summary of mean and median evaluation metrics on the Norman et al. dataset. The top-performing values are bolded and the second-best are underlined for each metric.

Model	Median R^2 (large)	Mean R^2 (large)	Median R^2 (DEG)	Mean R^2 (DEG)
KNN	0.982	0.942	0.986	0.948
Linear	-22.932	-27.767	-24.400	-27.518
Mean	-0.473	-0.743	-0.806	-0.650
Median	0.022	-0.006	-0.621	-0.691
PerturbNet	0.870	0.814	0.882	0.823
Training sample	0.697	0.647	0.852	0.787

Table 5: Summary of mean and median evaluation metrics on the Ursu et al. dataset. The top-performing values are bolded and the second-best are underlined for each metric.

Model	Median R^2 (large)	Mean R^2 (large)	Median R^2 (DEG)	Mean R^2 (DEG)
KNN	-4.031	-41.873	0.971	0.955
Linear	-10.186	-139.809	0.837	0.685
Mean	-68.699	-653.178	0.027	-0.208
Median	NA	NA	NA	NA
PerturbNet	0.2	-99.65	0.987	0.983
Training sample	-4.519	-44.121	0.972	0.958

Fig. 2 PerturbNet predicts response to small molecule treatment. (a) Diagram of the chemical variational autoencoder (ChemicalVAE) architecture for encoding small molecules represented as SMILES strings. The network was trained on the ZINC dataset, consisting of approximately 250,000 drug-like molecules. No gene expression information is used for training the drug representation network. Note that a cell representation network is also trained on gene expression values (not shown here). (b) Visualization of PerturbNet predictions for two distinct perturbations from the LINCS-Drug dataset. The UMAP coordinates are computed from the latent spaces of the perturbation network (left) and the cell state network (right). The mapping function learned by the cINN is indicated with lines connecting the perturbation and cell state representations. The predicted cell state distributions are also indicated with contour lines. (c) Box plots of R^2 calculated on all genes and the top 50 differentially expressed genes (DEGs) in one test split of the LINCS-Drug dataset for chemCPA and PerturbNet with (-iso) and without stereoisomers shared between train and

test sets. **(d)** Heatmap showing the proportion of train/test splits in which PerturbNet or ChemCPA (trained with stereoisomers) outperforms the same model trained without stereoisomers across five test splits of the LINCS-Drug dataset. "Outperforms" here means a significant difference by a one-sided Wilcoxon test. **(e)** Box plots of evaluation metrics in one test split of the LINCS-Drug dataset (plots for additional splits are shown in the supplement). **(f)** Box plots of evaluation metrics in one test split of the sci-Plex dataset (plots for additional splits are shown in the supplement). Some models are not displayed due to out-of-scale values for certain metrics. **(g)** Heatmap showing the proportion of train/test splits in which PerturbNet achieves significantly better performance (based on one-sided Wilcoxon tests) compared to competing models (columns) across five train/test splits of the sci-Plex and LINCS-Drug datasets.

Fig. 3 PerturbNet predicts response to genetic perturbation. **(a)** Diagram of the GenotypeVAE architecture. Perturbations are represented in terms of their gene ontology annotations. The network is trained on all one- and two-gene combinations (approximately 177 million). **(b)** Box plots of R^2 values for unseen genetic perturbations, calculated on all genes and the top 50 differentially expressed genes (DEGs) in one test split of the Norman et al. dataset (plots for additional splits are in the supplement). "PerturbNet-raw-count" utilizes a zero-inflated negative binomial (ZINB) likelihood for the cell representation network, while "PerturbNet-normalized" employs a Gaussian likelihood. **(c)** Box plots of R^2 values for unseen genetic perturbations, calculated on all genes and the top 50 DEGs in one test split of the Norman et al. dataset. **(d)** Heatmap showing the fraction of train/test splits in which PerturbNet achieves significantly better performance (based on one-sided Wilcoxon tests) compared to competing models (columns) on the Norman et al. dataset. **(e)–(f)** Scatter plots of R^2 values for unseen genetic perturbations, calculated across all genes, the top 50 differentially expressed genes (DEGs), and large-effect genes from the Norman et al. dataset. Data points are aggregated from all five test splits. Different colors represent the "number of seen genes". Labels such as "0/1" indicate that the test perturbation affects one unseen

gene, while "0/2" indicates two unseen genes are perturbed. "1/2" denotes that two genes are perturbed, but one of the target effects has already been observed independently or in combination with other genetic perturbations.

Fig. 4 PerturbNet predicts response to coding sequence mutation. (a) Diagram of the approach for training the representation network for coding sequence mutations. Each perturbation is an amino acid sequence edited by CRISPR. We used an evolutionary scale modeling (ESM) transformer pre-trained on the UniParc database, which contains 250 million sequences. (b) Workflow for predicting the effects of all possible point coding variants along a specific protein sequence. PerturbNet is trained using all available coding variants, and then used to generate cell state distributions for every possible protein variant at each position within the sequence. (c) Bar plots showing the cell type proportions (left) and the mean volume changes induced by coding variants (right) for the three predicted variant clusters along the GATA1 sequences. (d) Box plots of R^2 values for unseen genetic perturbations (coding variants), calculated on large-effect genes in the datasets from Ursu et al. (TP53 + KRAS) and Jorge et al. (GATA1). The median model is not shown for Jorge et al. because it produces out-of-range values. (e) Heatmap showing the proportion of train/test splits for which PerturbNet significantly outperforms (based on one-sided Wilcoxon tests) other models (columns). (f) UMAP visualization of predicted cell distributions for all possible point coding variants and observed coding variants on the GATA1 protein, with three distinct perturbation clusters highlighted. (g) Cell density of each perturbation cluster in the UMAP. (h) The top 10 predicted large-effect coding variants are located at or near critical GATA1-DNA interacting regions.

- Insufficient technical details on methodology: While there has been some improvement, the technical descriptions remain vague. Several methodological statements are ambiguous, preventing reproducibility. For example, "Data were preprocessed using Scanpy" lacks details on the specific preprocessing steps, normalization, or parameter choices. "Test set perturbations were at least partially unseen during training" is imprecise and requires clarification on the extent to which perturbations were unseen and how this was ensured. The description of model training, hyperparameter selection, and evaluation criteria is still too high-level and should be expanded with explicit details.

We have added these details to the Methods section, including the specific parameters used in Scanpy and the steps taken to create the split for the Norman et al. dataset. In detail, we add the code we used to preprocess the dataset in the methods:

```
sc.pp.normalize_total(adata, target_sum = 1e4)

sc.pp.log1p(adata)

sc.pp.highly_variable_genes(adata, min_mean=0.0125, max_mean=5, min_disp=0.5)
```

We also clarify the steps of creating test splits in Norman dataset: "To construct our test set, we randomly selected three different types of perturbations: (1) one gene perturbed and one gene unobserved; (2) two genes perturbed and one gene unobserved; and (3) two genes perturbed and two genes unobserved. This ensures that all test perturbations contain one or more unobserved gene targets."

For model training and hyperparameter selection, in addition to improving the original text descriptions, we also created tables to summarize the parameters for each model, which are shown at the bottom of this section.

For evaluation criteria, we improved the text descriptions and added formulas for each metric. We also included the scanpy function calls used to select differentially expressed genes.

Component	Configuration
Input	Shape: (batch_size, 120, 35)
Conv1D Layer 1	Conv1d(in=120, out=9, kernel_size=9) + Tanh + BatchNorm1d
Conv1D Layer 2	Conv1d(in=9, out=9, kernel_size=9) + Tanh + BatchNorm1d
Conv1D Layer 3	Conv1d(in=9, out=10, kernel_size=11) + Tanh + BatchNorm1d
Flatten + FC (Encoder)	Linear(90 → 196) + Tanh + Dropout + BatchNorm1d
Latent Layers	Two Linear(196 → 196) layers for mean and log-variance
Decoder FC	Linear(196 → 196) + Tanh + Dropout + BatchNorm1d
Repeat Vector	Expand to shape (batch_size, 120, 196)
GRU Decoder	GRU(input_size=196, hidden_size=488, num_layers=3) + Tanh
Final FC + Output	Linear(488 → 35) + Softmax (applied per timestep)
Output Shape	(batch_size, 120, 35)
Dropout Rate	0.0828
Latent Dim (z_dim)	196
GRU Hidden Size	488
Training Batch Size	128
Learning Rate	1×10^{-4}
Epochs	525
Dataset	ZINC

Table. S2: ChemicalVAE Model Architecture and Training Parameters

Component	Configuration
Input	Shape: (batch_size, 15,988); one-hot annotation vectors from GO
Encoder Layer 1	Linear(15988 → 512) + BatchNorm1d + LeakyReLU + Dropout
Encoder Layer 2	Linear(512 → 256) + BatchNorm1d + LeakyReLU + Dropout
Latent Mean Layer	Linear(256 → 10)
Latent Std Layer	Linear(256 → 10)
Decoder Layer 1	Linear(10 → 256) + BatchNorm1d + LeakyReLU + Dropout(p=0.2)
Decoder Layer 2	Linear(256 → 512) + BatchNorm1d + LeakyReLU + Dropout(p=0.2)
Output Layer	Linear(512 → 15988) + Sigmoid
Dropout Rate	0.2
Latent Dim (z_dim)	10
Training Batch Size	128
Learning Rate	1×10^{-4}
Epochs	300
Dataset	GO Consortium gene ontology annotations (single and double target genes)

Table. S3: GenotypeVAE Model Architecture and Training Parameters

Component	Configuration
Standard VAE (for normalized data)	
Input	Gene expression vectors (normalized data), shape: (batch_size, x_{dim})
Encoder Layer 1	Linear($x_{dim} \rightarrow 512$) + BatchNorm + LeakyReLU + Dropout(p=0.2)
Encoder Layer 2	Linear(512 \rightarrow 256) + BatchNorm + ReLU + Dropout(p=0.2)
Latent Mean Layer	Linear(256 \rightarrow 10)
Latent Scale Layer	Linear(256 \rightarrow 10) + Softplus
Decoder Layer 1	Linear(10 \rightarrow 256) + BatchNorm + LeakyReLU + Dropout
Decoder Layer 2	Linear(256 \rightarrow 512) + BatchNorm + LeakyReLU + Dropout
Output Layer	Linear(512 $\rightarrow x_{dim}$)
Latent Dim (z_{dim})	10
Dropout Rate	0.2
Learning Rate	1×10^{-4}
Batch Size	128
Epochs	150
scVI Model (for count data, using ZINB likelihood)	
Model Type	Variational Autoencoder from <code>scvi-tools 0.7.1</code> [32]
Likelihood	Zero-inflated Negative Binomial (ZINB)
Latent Dim (z_{dim})	10
Epochs	700 (default settings)
Sampling Strategy	Library size sampled from training set during generation

Table. S4: Cell Representation Network Architecture and Training Parameters

Component	Configuration
Flow Architecture	20 invertible blocks, each with:  - Alternating affine coupling layer - ActNorm layer - Fixed permutation layer
Embedding Module	 - Input: conditioning vector - 2 hidden layers, hidden dim = 256 - Output dim = 10 - Activation: LeakyReLU - Optional BatchNorm (enabled)
Shared Model Parameters	<code>in_channels = 10, embedding_dim = 10, hidden_dim = 1024,</code> <code>hidden_depth = 2, activation = none, conditioner.use_bn =</code> <code>True</code>
Dataset-specific Configuration	
Norman et al.	<code>conditioning_dim = 10, epochs = 50</code>
Ursu et al.	<code>conditioning_dim = 1280, epochs = 50</code>
Jorge et al.	<code>conditioning_dim = 1280, epochs = 50</code>
LINCS-Drug	<code>conditioning_dim = 196, epochs = 100</code>
sci-Plex	<code>conditioning_dim = 200, epochs = 100</code>
Training Parameters	Batch size = 128, Learning rate = 4.5×10^{-6}

Table. S5: Conditional Invertible Neural Network (cINN) Architecture and Training Parameters

- Generalization of the generated profiles: Thank you for the clarification on how the sampling is performed. In the notebooks provided (https://github.com/welch-lab/PerturbNet/blob/main/notebooks/Tutorial_PerturbNet_coding_variants.ipynb, https://github.com/welch-lab/PerturbNet/blob/main/notebooks/Tutorial_PerturbNet_Genetic.ipynb) it is possible to see areas of real data (in the UMAP plots) which are not well covered by the generated profiles. Investigating these could give us a better understanding of the limitations of the method.

The UMAPs of predicted and observed data are in fact very well-mixed, indicating that PerturbNet is accurately modeling the data. See these plots:

Umap in genetic perturbation notebook

Umap in coding variants perturbation notebook

Perhaps the reviewer is referring to the plot showing different perturbation effects (such as the one below). This plot is supposed to show different distributions to highlight perturbation-induced cell state shifts. The real cells (shown in gray on the background) represent the full spectrum of cell states in the dataset, while an individual perturbation (blue or red in this figure) occupies a subset of the cell state landscape.

Example visualization of PerturbNet predictions for two distinct perturbations from the LINCS-Drug dataset.

- Organization and writing: The manuscript remains poorly structured, with key methodological details dispersed in a way that makes comprehension challenging. The flow of information should be improved to guide the reader more effectively through the motivation, methodology, results, and implications.

We modified the presentation of the methodology in the main text to summarize the architectures used in a single place and in a top-down fashion, referring to the methods section for further details. This makes it easier to parse the presentation of the subsequent result sections.

Reviewer #1 (Remarks on code availability):

Python 3.7 is no longer supported as of June 27, 2023. To maximize the adoption and maintainability of source code, I would recommended updating to a later version of Python (3.13 is the latest released version, but anything above 3.10 is still relatively

modern). Documentation of the programmatic interface beyond the notebooks would also provide a better way to increase adoption of the method.

We rewrote all of the tensorflow parts of the code in PyTorch, upgraded to a recent Python version, and added Sphinx autodocs. The ReadTheDocs page is here:

<https://perturbnet.readthedocs.io/en/latest/>

Reviewer 2

I think the authors have strengthened their case that this method is currently the state-of-the-art for perturbation prediction, which is proving to be a difficult problem. This includes addressing comparison to published methods and addressing some of the concerns raised by recent preprints that have highlighted that these models are often defeated by simple baselines. The paper is also quite easy to read (and in fact could probably be shortened without losing much).

One challenge is that the absolute improvements relative to simpler baselines are not that large. E.g., a k-nn model performs well in Figs. 2d and 6d and a linear model performs well in Fig. 3c, in line with the preprints. This is despite these models having no access to the side information implicit in the representations used in PerturbNet. Nevertheless, it does seem to outperform them most of the time, which does seem to move the needle in the field.

Given these modest improvements relative to baselines, it's not entirely clear what new capabilities PerturbNet unlocks in its current form. As the authors state, this may be partly due to the limited existing datasets for training, which I do suspect will improve quickly. A separate advantage is that the architecture here is simple and easily extendable in the future, so perhaps learning better representations or encoding more information at this step is a path to better predictive power. I do see and acknowledge the potential, and it's possible this architecture is the right direction to focus.

We would like to emphasize that we are the first to show that effects of protein-coding mutations on global gene expression can be predicted from the mutated protein sequence. This is an important qualitative contribution beyond the quantitative bump in performance that we show.

I think if the authors can truly establish that their method is the state-of-the-art then it is a valuable advance, as defeating simple baselines has proven difficult. This requires a handful of additional comparisons that I don't think are particularly onerous to carry out.

Otherwise, I think this work is suitable for a more technical journal as this field continues to develop.

Specific comments:

A recent preprint (A systematic comparison of computational methods for expression forecasting, <https://doi.org/10.1101/2023.07.28.551039>) introduced new baselines that outperform the linear one used here, including mean and median. It is also packaged in a github repository, which I think should make comparisons easy to perform: https://github.com/ekernf01/perturbation_benchmarking/. Can you please show improvement of PerturbNet relative to the mean/median baselines?

We have computed these additional baselines, and PerturbNet outperforms them.

Heatmap showing the proportion of train/test splits in which PerturbNet achieves significantly better performance (based on the one-sided Wilcoxon tests) compared to competing models (columns) across five train/test splits of (a) sci-Plex and LINC5-Drug dataset. (b) Norman et al. dataset. (c) Ursu et al. and Jorge et al. dataset.

Boxplots of evaluation metrics on the LINC-Drug and sci-Plex datasets. (a) Comparison between PerturbNet and ChemCPA trained with and without stereoisomers. (b) Benchmark results on the LINC-Drug dataset. (c) Benchmark results on the sci-Plex dataset.

Boxplots of evaluation metrics on the genetic perturbation datasets. (a) Comparison between PerturbNet trained on raw counts and on normalized expression. (b) Benchmark results on the Norman et al. dataset. (c) Benchmark results on the Ursu et al. and Jorge et al. datasets.

I am still concerned about Figure 4. As noted in my previous review, I don't have technical expertise to comment deeply here, but I just don't think it makes sense to decompose the function of small molecules atom by atom in this manner. What is the physical interpretation of these scores supposed to be? Most small molecules interact non-specifically with many binding partners. Are predictions for stereoisomers or structurally similar molecules concordant? I am wondering if the authors would be willing to discard or deemphasize this portion of the paper. Otherwise, I think it should be evaluated by a chemist. Alternatively, it seems like a similar case for interpretability

could instead be made with the data in Figure 6 where at least there is a structural interpretation.

We have significantly de-emphasized the discussion of the interpretability analysis and moved the main figure to a supplementary figure.

21st May 2025

Manuscript Number: MSB-2025-13097

Title: PerturbNet predicts single-cell responses to unseen chemical and genetic perturbations

Dear Dr Welch,

Thank you for the submission of your reviewed and revised manuscript to Molecular Systems Biology. As discussed, we consulted an expert advisor to assess whether the reviewer concerns had been sufficiently addressed and if they agreed that the manuscript was suitable for Molecular Systems Biology. I am pleased to inform you that the advisor was supportive, finding the overall goal of the manuscript and method very important, and we will be able to accept your manuscript pending the following final amendments:

1) Please download the EMBO Press "Author Checklist" and complete all relevant questions. This file should be uploaded with your submission. This file can be downloaded from our website at:

<https://www.embopress.org/page/journal/17444292/authorguide>

2) We note that currently Yuxuan Song is listed as an author in the manuscript but has not been included in the manuscript submission system as an author. Please double check this and ensure that both the manuscript and our submission system have the same authors listed.

3) Please upload the manuscript as a .docx formatted version of the manuscript text (including legends for main figures, EV figures and tables) with no track changes. Alternatively you may submit the manuscript in LaTeX format.

4) In the main manuscript file, please include keywords to max. 5.

5) Please remove the Code Availability section and include the information in the Data availability section, formatted according to the example below and only including newly generated code/datasets:

"The datasets and computer code produced in this study are available in the following databases:

- Chip-Seq data: Gene Expression Omnibus GSE46748 (<https://www.ncbi.nlm.nih.gov/geo/query/acc.cgi?acc=GSE46748>)

- Modeling computer scripts: GitHub (<https://github.com/SysBioChalmers/GECKO/releases/tag/v1.0>)

- [data type]: [full name of the resource] [accession number/identifier] ([doi or URL or identifiers.org/DATABASE:ACCESSION])"

6) Please rename "Competing Interests" to "Disclosure and competing interests statement". We updated our journal's competing interests policy in January 2022 and request authors to consider both actual and perceived competing interests. Please review the policy <https://www.embopress.org/competing-interests> and update your competing interests if necessary.

7) Author contributions: Please remove it from the manuscript and specify author contributions in our submission system.

CRedit has replaced the traditional author contributions section because it offers a systematic machine-readable author contributions format that allows for more effective research assessment. You are encouraged to use the free text boxes beneath each contributing author's name to add specific details on the author's contribution. More information is available in our guide to authors:

<https://www.embopress.org/page/journal/17574684/authorguide#authorshipguidelines>

8) References: Please correct the reference citation in the reference list to be alphabetical (not numerical). Where there are more than 10 authors on a paper, only the first 10 should be listed, followed by "et al.". Please check "Author Guidelines" for more information.

<https://www.embopress.org/page/journal/17574684/authorguide#referencesformat>

10) Data not shown: We do not allow statements/conclusions with "data not shown". As per our guidelines, on "Unpublished Data" the journal does not permit citation of "Data not shown". All data referred to in the paper should be displayed in the main or Expanded View figures. Please remove from pages 8 and 17.

11) All Materials and Methods need to be described in the main text using our 'Structured Methods' format. According to this format, the Methods section includes a Reagents and Tools Table (listing key reagents, experimental models, software and relevant equipment and including their sources and relevant identifiers) followed by a Methods and Protocols section describing the methods, ideally using a step-by-step protocol format. The aim is to facilitate adoption of the methodologies across labs. Please download and fill our Reagents and Tools Table template (.docx), which you can find in our author guidelines:

<https://www.embopress.org/doi/10.15252/msb.20178071>. "

12) Please place individual sections of the manuscript in the following order: Title page - Abstract & Keywords - Introduction - Results - Discussion - Methods - Data Availability - Acknowledgements - Disclosure and Competing Interests Statement - References - Figure Legends - Expanded View Figure Legends.

13) For the figures and figure legends, please take care of the following:

- Please remove all figures from main manuscript file and leave only main figure legends placed after the references. If you would like to do so, some of the supplementary figures can be made into Expanded View figures. In this case, each figure will still need to fit onto one page and be renamed as Figure EV1, etc. Please ensure that the callouts are also updated in the main manuscript. For EV figures the legends should stay in the manuscript, with the heading Expanded View Figures Legends, and placed after the main figure legends. The remaining supplementary figures can be compiled into the Appendix file (ensuring that the callouts are updated appropriately, see more below). Main figures and any EV figures should be uploaded as individual, high-resolution files. Please check "Author Guidelines" for more information:

<https://www.embopress.org/page/journal/17574684/authorguide#figureformat>

- Please note that the box plots need to be defined in terms of minima, maxima, centre, bounds of box and whiskers, and percentile in the legends of figures 2C, E, F; 3B, C; 4D

- Please note that information related to n is missing in the legends of figures 2C, E, F; 3B, C; 4D

14) Appendix file: Please upload the Appendix as a single PDF containing all of the supplementary figures and tables (no separate image files are needed) including a title page with "Appendix for + manuscript title" and a Table of Contents with page numbers. Please also ensure the word "Appendix" is included in all labels for Appendix Figures and Appendix Tables including in the Table of Contents and in the callouts in the main manuscript.

15) Funding: Please ensure that all funding sources included in the manuscript are entered the same into the manuscript submission system. Currently there is a mismatch between grant number R01HG101883 in the manuscript versus R01HG010883 in our submission system. In addition, U01HG011952 is missing from our submission system.

16) Synopsis:

- Synopsis image: Please provide a graphic that summarises the main findings of the manuscript on a glance and upload it as a high-resolution jpeg file 550 pixels wide x (300-600) pixels high.

- Synopsis text: Please provide a short standfirst (maximum of 300 characters, including space), limit the bullet points to max. 5 and upload it as a separate .doc file. Please write the bullet points to summarise the key NEW findings. They should be designed to be complementary to the abstract - i.e. not repeat the same text. We encourage inclusion of key acronyms and quantitative information (maximum of 30 words / bullet point). Please use the passive voice.

17) Source Data: Please ensure that a completed Source Data checklist is uploaded as a Related Manuscript File. Source Data should be organized as a single source data file (zipped) per figure for main figures (all EV and/or Appendix figure Source Data can be included in a single folder), with the panels clearly visible in the folder structure instead of a single excel file for all Source Data. e.g. all the Source data files for figure 1 need to be saved in a single folder and this needs to be zipped and then uploaded as "SD figure 1.zip" file.

18) As part of the EMBO Publications transparent editorial process initiative (see our policy here:

https://www.embopress.org/transparent-process#Review_Process), Molecular Systems Biology will publish online a Peer Review File (PRF) to accompany accepted manuscripts. This file will be published in conjunction with your paper and will include the anonymous referee reports, your point-by-point response and all pertinent correspondence relating to the manuscript. Let us know whether you agree with the publication of the PRF and as here, if you want to remove or not any figures from it prior to publication. Please note that the Authors checklist will be published at the end of the PRF.

19) After your paper is published, we may promote it on social media. If you have any handles or hashtags for Bluesky you would like included, please let us know.

20) Please provide a point-by-point letter INCLUDING my comments and your detailed responses (as Word file).

I look forward to reading a new revised version of your manuscript as soon as possible.

Yours sincerely,

Poonam Bheda, PhD
Scientific Editor
Molecular Systems Biology

If you do choose to resubmit, please click on the link below to submit the revision online before 20th Jun 2025.

IMPORTANT: When you send your revision, we will require the following items:

1. the manuscript text in LaTeX, RTF or MS Word format

2. a letter with a detailed description of the changes made in response to the referees. Please specify clearly the exact places in the text (pages and paragraphs) where each change has been made in response to each specific comment given
3. three to four 'bullet points' highlighting the main findings of your study
4. a short 'blurb' text summarizing in two sentences the study (max. 250 characters)
5. a 'thumbnail image' (550px width and max 400px height, Illustrator, PowerPoint or jpeg format), which can be used as 'visual title' for the synopsis section of your paper.
6. Please include an author contributions statement after the Acknowledgements section (see <https://www.embopress.org/page/journal/17444292/authorguide#manuscriptpreparation>)
7. Please complete the CHECKLIST available at (<https://bit.ly/EMBOPressAuthorChecklist>). Please note that the Author Checklist will be published alongside the paper as part of the transparent process (<https://www.embopress.org/page/journal/17444292/authorguide#transparentprocess>).
8. When assembling figures, please refer to our figure preparation guideline in order to ensure proper formatting and readability in print as well as on screen:
<https://bit.ly/EMBOPressFigurePreparationGuideline>
See also figure legend guidelines: <https://www.embopress.org/page/journal/17444292/authorguide#figureformat>
9. Please note that corresponding authors are required to supply an ORCID ID for their name upon submission of a revised manuscript (EMBO Press signed a joint statement to encourage ORCID adoption). (<https://www.embopress.org/page/journal/17444292/authorguide#editorialprocess>)
Currently, our records indicate that the ORCID for your account is 0000-0002-5869-2391.

Link Not Available

10. Include a Reagents and Tools Table as part of the Methods section, which can be downloaded from our author guidelines (<https://www.embopress.org/page/journal/17444292/authorguide#structuredmethods>)

*** PLEASE NOTE *** As part of the EMBO Press transparent editorial process initiative (see our Editorial at <https://dx.doi.org/10.1038/msb.2010.72> , Molecular Systems Biology will publish online a Review Process File to accompany accepted manuscripts. When preparing your letter of response, please be aware that in the event of acceptance, your cover letter/point-by-point document will be included as part of this File, which will be available to the scientific community. More information about this initiative is available in our Instructions to Authors. If you have any questions about this initiative, please contact the editorial office (msb@embo.org).

1) Please download the EMBO Press "Author Checklist" and complete all relevant questions. This file should be uploaded with your submission. This file can be downloaded from our website at: <https://www.embopress.org/page/journal/17444292/authorguide>

2) We note that currently Yuxuan Song is listed as an author in the manuscript but has not been included in the manuscript submission system as an author. Please double check this and ensure that both the manuscript and our submission system have the same authors listed. **We corrected this by including Yuxuan Song in the submission system.**

3) Please upload the manuscript as a .docx formatted version of the manuscript text (including legends for main figures, EV figures and tables) with no track changes. Alternatively you may submit the manuscript in LaTeX format. **We uploaded the LaTeX version of the manuscript.**

4) In the main manuscript file, please include keywords to max. 5. **According to the EMBO guidelines (<https://www.embopress.org/page/journal/17444292/authorguide>) we have added 5 keywords and separated them by "/". They are: deep generative models / genome editing / high-throughput screening / perturbation prediction / single-cell transcriptomics.**

5) Please remove the Code Availability section and include the information in the Data availability section, formatted according to the example below and only including newly generated code/datasets:

"The datasets and computer code produced in this study are available in the following databases:

- Chip-Seq data: Gene Expression Omnibus GSE46748

(<https://www.ncbi.nlm.nih.gov/geo/query/acc.cgi?acc=GSE46748>)

- Modeling computer scripts: GitHub

(<https://github.com/SysBioChalmers/GECKO/releases/tag/v1.0>)

- [data type]: [full name of the resource] [accession number/identifier] ([doi or URL or identifiers.org/DATABASE:ACCESSION)]"

We have merged sections and reformatted to align with the example.

6) Please rename "Competing Interests" to "Disclosure and competing interests statement". We updated our journal's competing interests policy in January 2022 and request authors to consider both actual and perceived competing interests. Please review the policy <https://www.embopress.org/competing-interests> and update your competing interests if necessary.

We have renamed the section. The competing interests statement stays the same.

7) Author contributions: Please remove it from the manuscript and specify author contributions in our submission system. CRediT has replaced the traditional author contributions section

because it offers a systematic machine-readable author contributions format that allows for more effective research assessment. You are encouraged to use the free text boxes beneath each contributing author's name to add specific details on the author's contribution. More information is available in our guide to authors:

<https://www.embopress.org/page/journal/17574684/authorguide#authorshipguidelines>

We have removed the section and specified the author contributions in the online manuscript system.

8) References: Please correct the reference citation in the reference list to be alphabetical (not numerical). Where there are more than 10 authors on a paper, only the first 10 should be listed, followed by "et al.". Please check "Author Guidelines" for more information.

<https://www.embopress.org/page/journal/17574684/authorguide#referencesformat>

We have corrected the reference citations to be alphabetical in "author + year" format.

9) Our journal encourages inclusion of *data citations in the reference list* to directly cite datasets that were re-used and obtained from public databases. Data citations in the article text are distinct from normal bibliographical citations and should directly link to the database records from which the data can be accessed. In the main text, data citations are formatted as follows: "Data ref: Smith et al, 2001" or "Data ref: NCBI Sequence Read Archive PRJNA342805, 2017". In the Reference list, data citations must be labeled with "[DATASET]". A data reference must provide the database name, accession number/identifiers and a resolvable link to the landing page from which the data can be accessed at the end of the reference. Further instructions are available at <https://www.embopress.org/page/journal/17574684/authorguide#referencesformat>.

We have updated the reference list and labeled them following the guidelines. The preprint is also labeled with [PREPRINT].

10) Data not shown: We do not allow statements/conclusions with "data not shown". As per our guidelines, on "Unpublished Data" the journal does not permit citation of "Data not shown". All data referred to in the paper should be displayed in the main or Expanded View figures. Please remove from pages 8 and 17.

Thank you for pointing that out. The data were actually used when generating the figure; we were simply pointing out that all of the values for some samples fall outside the axis limits. We have rephrased the figure legends to avoid the impression that the data are not shown. For the figures from page 8 and page 17, we replaced the legend with "The y axis is truncated at 0." We have also included these data in our source data files.

11) All Materials and Methods need to be described in the main text using our 'Structured Methods' format. According to this format, the Methods section includes a Reagents and Tools Table (listing key reagents, experimental models, software and relevant equipment and including their sources and relevant identifiers) followed by a Methods and Protocols section

describing the methods, ideally using a step-by-step protocol format. The aim is to facilitate adoption of the methodologies across labs.

Please download and fill our Reagents and Tools Table template (.docx), which you can find in our author guidelines:

<https://www.embopress.org/doi/10.15252/msb.20178071>. "

We have filled the form and uploaded it as a separate .docx file.

12) Please place individual sections of the manuscript in the following order: Title page - Abstract & Keywords - Introduction - Results - Discussion - Methods - Data Availability - Acknowledgements - Disclosure and Competing Interests Statement - References - Figure Legends - Expanded View Figure Legends.

We have adjusted the order of sections according to the order you commented. According to the author guidelines we also need to move the main tables to the bottom of the manuscript. So we created an extra "Tables" section after "Figure Legends" to present the main tables.

13) For the figures and figure legends, please take care of the following:

- Please remove all figures from main manuscript file and leave only main figure legends placed after the references. If you would like to do so, some of the supplementary figures can be made into Expanded View figures. In this case, each figure will still need to fit onto one page and be renamed as Figure EV1, etc. Please ensure that the callouts are also updated in the main manuscript. For EV figures the legends should stay in the manuscript, with the heading Expanded View Figures Legends, and placed after the main figure legends. The remaining supplementary figures can be compiled into the Appendix file (ensuring that the callouts are updated appropriately, see more below). Main figures and any EV figures should be uploaded as individual, high-resolution files. Please check "Author Guidelines" for more information:

<https://www.embopress.org/page/journal/17574684/authorguide#figureformat>

- Please note that the box plots need to be defined in terms of minima, maxima, centre, bounds of box and whiskers, and percentile in the legends of figures 2C, E, F; 3B, C; 4D

- Please note that information related to n is missing in the legends of figures 2C, E, F; 3B, C; 4D

We have added the sentences below to the figure legends for Fig. 2, 3, and 4 to define the box plots: "for all box plots in this panel, the box plots show the median (center line), the 25th and 75th percentiles (box bounds), and 1.5× the interquartile range (whiskers). Points beyond the whiskers are plotted as outliers."

We have added sample size n for each boxplot in the legends.

For each figure, we also changed the labels to upper case according to the author guidelines.

14) Appendix file: Please upload the Appendix as a single PDF containing all of the supplementary figures and tables (no separate image files are needed) including a title page with "Appendix for + manuscript title" and a Table of Contents with page numbers. Please also ensure the word "Appendix" is included in all labels for Appendix Figures and Appendix Tables including in the Table of Contents and in the callouts in the main manuscript.

We have added the word "Appendix" for all of the supplementary figures and tables and in the callouts in the main manuscript. We have separated the appendix to the main text and added the tables of content contents with page numbers.

15) Funding: Please ensure that all funding sources included in the manuscript are entered the same into the manuscript submission system. Currently there is a mismatch between grant number R01HG101883 in the manuscript versus R01HG010883 in our submission system. In addition, U01HG011952 is missing from our submission system.

We have updated the funding listed in the paper and submission system. The grants are now correctly cited.

16) Synopsis:

- Synopsis image: Please provide a graphic that summarises the main findings of the manuscript on a glance and upload it as a high-resolution jpeg file 550 pixels wide x (300-600) pixels high.
- Synopsis text: Please provide a short standfirst (maximum of 300 characters, including space), limit the bullet points to max. 5 and upload it as a separate .doc file. Please write the bullet points to summarise the key NEW findings. They should be designed to be complementary to the abstract - i.e. not repeat the same text. We encourage inclusion of key acronyms and quantitative information (maximum of 30 words / bullet point). Please use the passive voice.

PerturbNet is a generative AI model that can predict shifts in cell state—changes in overall gene expression—in response to multiple types of unseen cellular perturbations.

- PerturbNet uses a flexible framework to “mix and match” neural networks and predict effects of chemical, gene knockdown, gene overexpression, and DNA sequence mutations.
- Despite being an all-in-one model, PerturbNet shows competitive performance with previous methods tailored to predict either chemical or genetic effects
- PerturbNet accurately predicts gene expression changes induced by coding sequence mutations in TP53, KRAS, and GATA1.
- An *in silico* screen of all possible single amino acid substitutions in GATA1 identifies candidates likely to modify erythroid cell differentiation.

17) Source Data: Please ensure that a completed Source Data checklist is uploaded as a Related Manuscript File. Source Data should be organized as a single source data file (zipped) per figure for main figures (all EV and/or Appendix figure Source Data can be included in a single folder), with the panels clearly visible in the folder structure instead of a single excel file for all Source Data. e.g. all the Source data files for figure 1 need to be saved in a single folder and this needs to be zipped and then uploaded as "SD figure 1.zip" file.

18) As part of the EMBO Publications transparent editorial process initiative (see our policy here: https://www.embopress.org/transparent-process#Review_Process), Molecular Systems Biology will publish online a Peer Review File (PRF) to accompany accepted manuscripts. This file will be published in conjunction with your paper and will include the anonymous referee reports, your point-by-point response and all pertinent correspondence relating to the manuscript. Let us know whether you agree with the publication of the PRF and as here, if you want to remove or not any figures from it prior to publication. Please note that the Authors checklist will be published at the end of the PRF.

We agree with the publication of the PRF. No changes are needed.

19) After your paper is published, we may promote it on social media. If you have any handles or hashtags for Bluesky you would like included, please let us know.

Twitter handles: @LabWelch, @WeizhouQ18748

LinkedIn profiles: <https://www.linkedin.com/in/joshua-welch-12360790/>

<https://www.linkedin.com/in/weizhou-qian-a81398246/>

20) Please provide a point-by-point letter INCLUDING my comments and your detailed responses (as Word file).

17th Jun 2025

Manuscript number: MSB-2025-13097R

Title: PerturbNet predicts single-cell responses to unseen chemical and genetic perturbations

Dear Dr Welch,

Thank you again for sending us your revised manuscript. We are now satisfied with the modifications made and I am pleased to inform you that your paper has been accepted for publication.

Yours sincerely,

Sincerely,

Poonam Bheda, PhD
Scientific Editor
Molecular Systems Biology
